# A brainstem monosynaptic excitatory pathway that drives locomotor activities and sympathetic cardiovascular responses

Satoshi Koba [1] ✉, Nao Kumada[1,2], Emi Narai[1], Naoya Kataoka [3,4], Kazuhiro Nakamura [3] & Tatsuo Watanabe[1]

Exercise including locomotion requires appropriate autonomic cardiovascular adjustments to meet the metabolic demands of contracting muscles, yet the functional brain architecture underlying these adjustments remains unknown. Here, we demonstrate brainstem circuitry that plays an essential role in relaying volitional motor signals, i.e., central command, to drive locomotor activities and sympathetic cardiovascular responses. Mesencephalic loco-motor neurons in rats transmit central command-driven excitatory signals onto the rostral ventrolateral medulla at least partially via glutamatergic processes, to activate both somatomotor and sympathetic nervous systems. Optogenetic excitation of this monosynaptic pathway elicits locomotor and cardiovascular responses as seen during running exercise, whereas pathway inhibition suppresses the locomotor activities and blood pressure elevation during voluntary running without affecting basal cardiovascular homeostasis. These results demonstrate an important subcortical pathway that transmits central command signals, providing a key insight into the central circuit mechanism required for the physiological conditioning essential to maximize exercise performance.

Exercise including locomotion, which is part of fundamental behavior in vertebrates including humans, is accompanied by autonomic cardiovascular adjustments that provide the metabolic resources, such as fuel and oxygen, demanded by contracting skeletal muscles and thereby boost physical performance. The contribution of a feedforward descending motor signal from the forebrain to the cardiovascular control has been suggested for more than a century[1,2]. Currently, this feedforward signal has been called central command and postulated as parallel activation of the somatic and autonomic motor systems in the brain to simultaneously increase muscle activity along with arterial pressure and cardiac contractility[3]. This concept first came from a human study showing that the magnitude of cardiovascular responses during voluntary isometric exercise at constant muscle tension positively correlated with the amount of central command activation that

was changed by reflexive contractions due to tendon vibration on either agonist or antagonist muscle[4]. Central command is coupled to activation of the sympathetic nervous system independently of movement feedback as shown by increases in cardiovascular variables during fictive locomotion in decorticate, paralyzed cats[5] and by exaggerated cardiovascular responses to voluntary muscle contraction in human subjects after paralysis[6,7].

The precise location of the source of central command remains unclear because the central circuit mechanism by which central command signals elicit autonomic cardiovascular adjustments during exercise has yet to be fully elucidated. Autonomic brain regions activated in response to voluntary exercise[8–12] or regions of which stimulation elicits either autonomic or somatomotor responses[13–18] may be involved in the central command control of circulation. For example,

[1]Division of Integrative Physiology, Tottori University Faculty of Medicine, Yonago, Japan. [2]Division of Integrative Bioscience, Tottori University Graduate School of Medical Sciences, Yonago, Japan. [3]Department of Integrative Physiology, Nagoya University Graduate School of Medicine, Nagoya, Japan. [4]Nagoya University Institute for Advanced Research, Nagoya, Japan. ✉e-mail: skoba@tottori-u.ac.jp

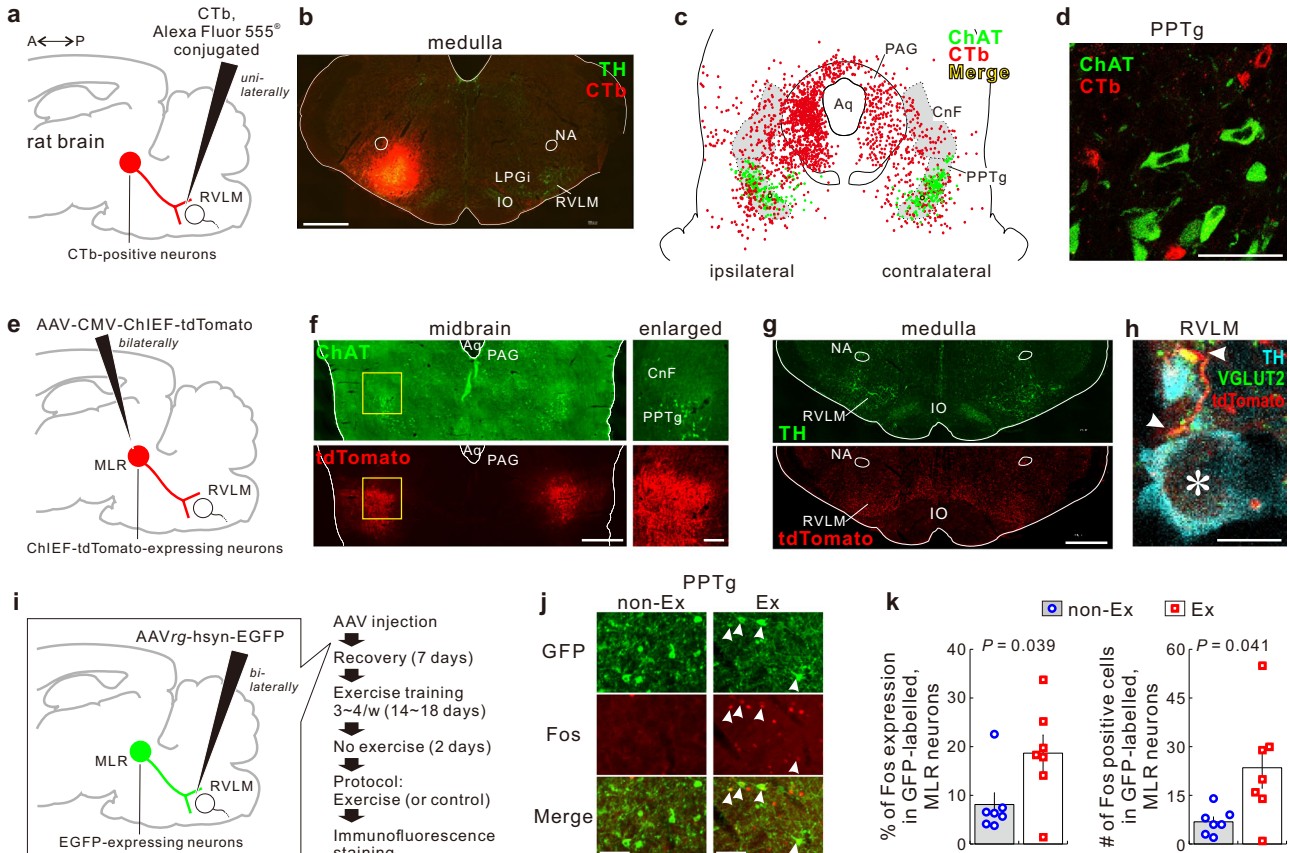

**Fig. 1 | MLR neurons projecting to the RVLM mediates central command signaling. a, b** CTb injection into the RVLM unilaterally. TH tyrosine hydroxylase; IO inferior olive nucleus; LPGi lateral paragigantocellular nucleus; NA nucleus anbiguus. Scale bar: 1 mm. **c** Drawing of the distribution of CTb-labeled cells [combined from 8 rats (6 males, 2 females)] and CTb-positive or -negative ChAT-immunoreactive cells [combined from 3 of 8 rats (2 males and 1 female)] in the midbrain coronal section. Aq aqueduct; PAG periaqueductal gray. **d** Confocal image showing that CTb-labeled PPTg cells were not merged with ChAT-immunoreactive cells. Scale bar: 50 μm. **e, f** AAV-CMV-ChIEF-tdTomato injections into the MLR bilaterally. Scale bars: 1 mm (midbrain) and 200 μm (enlarged). **g** tdTomato-labeled axons distributed in the ventral medulla. Scale bar: 1 mm. **h** Confocal image showing close associations of tdTomato-labeled axonal swellings containing VGLUT2 (arrowheads) with RVLM C1 neuronal dendrite and cell body

(asterisk). Scale bar: 10 μm. **i** Experimental schema to study Fos immunoreactivities in RVLM-projecting MLR neurons of treadmill-exercised and non-exercised rats. **j** Immunofluorescence staining of GFP and Fos. Arrowheads indicate Fos expression in GFP-labeled cells found in the PPTg. Scale bars: 100 μm. **k** Comparisons of Fos-immunoreactive cells in GFP-labeled, RVLM-projecting MLR neurons between non-exercised controls and exercised rats (*n* = 7 for each distinct group; all males). Data were analyzed by two-sided Welch's *t*-test; statistic information including *t* statistic values and degrees of freedom is presented in Supplementary Table 15. Data shown are means ± SEM. Source data are provided as a Source Data file. The brain section images used in figures (**a, e, i**) were adapted from "Paxinos G & Watson C. The rat brain in stereotaxic coordinates. 6th edn. (Amsterdam, Academic Press/Elsevier, 2007)".

human studies using neurosurgical techniques suggested that mesencephalic circuits including the subthalamic nucleus (STN) and periaqueductal gray (PAG), whose neuronal activities are elevated during volitional exercise[11], configure subcortical circuitry to relay central command signals[19] as evidenced by the pressor effect of electrical stimulation of the STN[17] or dorsal/lateral PAG[18] in awake patients with Parkinson's disease or chronic pain. Nonetheless, causal roles of these brain regions in autonomic changes during exercise as well as functional connections with other regions have not been demonstrated. The brain substrate of central command has also gained clinical importance. Abnormal cardiovascular regulation during exercise in pathological conditions, such as heart failure, increases exercise intolerance and the risk of fatal cardiac events such as arrhythmia[20]. This is at least partly caused by central command dysfunction[21,22], whereas therapeutic exercise programs for patients improve their functional status and outcome[23].

The rostral ventrolateral medulla (RVLM), which is located immediately caudal to the caudal pole of the facial nucleus, contains spinally projecting, sympathetic premotor neurons, more than half of which are adrenergic C1 neurons[24]. Both RVLM C1 and non-C1 neurons

are excited by voluntary exercise[8,9,12] and play a pivotal role in regulating sympathetic vasomotor tone[25–28]. Thus, the RVLM is likely a key node of the central circuit mechanism for sympathetic cardiovascular responses during exercise[29]. In the present study, to identify a subcortical monosynaptic pathway that transmits central command signals to the RVLM during voluntary exercise to elicit cardiovascular responses, we performed functional neuroanatomy and in vivo physiological experiments in rats to record peripheral nerve discharges, cardiovascular changes, and behaviors in combination with optogenetic techniques to manipulate a pathway of interest.

## Results

### RVLM receives glutamatergic projections from the mesencephalic locomotor region

We first sought for RVLM-projecting neurons in rats via retrograde tracing with the tracer, cholera toxin b subunit (CTb), unilaterally injected into the RVLM (Fig. 1a, b). A significant number of CTb-labeled neuronal cell bodies were found to be bilaterally distributed in midbrain areas including the cuneiform nucleus (CnF) and pedunculopontine tegmental nucleus (PPTg), the latter of which harbored many

cholinergic neurons (Fig. 1c). These two nuclei constitute a functional area known as the mesencephalic locomotor region (MLR), which is capable of initiating and controlling locomotion and is evolutionarily well conserved across mammalian species including humans[30,31]. Therefore, we focused on this monosynaptic MLR → RVLM pathway as a candidate for the subcortical route that mediates the central command signaling engaged for locomotor exercise. The MLR contains glutamatergic and GABAergic neurons, besides the cholinergic PPTg neurons, which bilaterally send numerous projections throughout the medullary reticular formation[32]. However, very few CTb-labeled PPTg neurons were immunoreactive for choline acetyltransferase (ChAT) (0.6 ± 0.3%, $n = 3$) (Fig. 1d). Thus, RVLM-projecting MLR neurons are principally noncholinergic.

To anterogradely trace the MLR → RVLM pathway, we made bilateral injections into the region across the CnF and PPTg with an adeno-associated virus vector (AAV) encoding ChIEF, a channelrhodopsin variant, fused with tdTomato (Fig. 1e, f). ChIEF-tdTomato-labeled, MLR-derived axons were abundantly distributed in the ventral part of the medulla (Fig. 1g); moreover, the MLR-derived axonal swellings that contained vesicular glutamate transporter 2 (VGLUT2) were closely associated with cell bodies and dendrites of RVLM C1 neurons (Fig. 1h), suggesting that synaptic contact exists between glutamatergic MLR → RVLM neurons and RVLM C1 neurons. Overall, these observations indicate that the RVLM receives monosynaptic glutamatergic bilateral projections from the MLR.

### Voluntary running activates MLR → RVLM neurons

We next questioned whether the MLR → RVLM pathway is activated by running exercise. We investigated expression of Fos, a biochemical correlate of increased firing, in MLR → RVLM projection neurons following voluntary running in rats that had received bilateral RVLM injections of a retrograde AAV (AAVrg) encoding EGFP; these rats had been trained to become accustomed to voluntary treadmill exercise (Fig. 1i). In agreement with the observations from CTb-injected rats, AAVrg transduction of RVLM-projecting neurons resulted in the dense localization of EGFP-labeled, noncholinergic cell bodies in the MLR across the CnF and PPTg (Supplementary Fig. 1a, b). Voluntary running on the treadmill (16 m/min for 40 min) increased Fos expression in EGFP-labeled (i.e., RVLM-projecting) MLR neurons, compared to non-exercise control (Fig. 1j, k). These results indicate that voluntary running exercise activates the MLR → RVLM pathway, consistent with the notion that this monosynaptic pathway is a part of the central circuit mechanism that mediates the volitional central command signaling engaged for running exercise.

Increased Fos expression after treadmill exercise was also observed in ChAT-immunoreactive PPTg neurons (Supplementary Fig. 1c, d). Excited cholinergic PPTg neurons may play a role in accelerating, but not eliciting, locomotion during running exercise[33].

### Excited MLR → RVLM neurons drive sympathoexcitation

To study the role of MLR → RVLM neurons in autonomic cardiovascular regulation, we examined the effect of optogenetic stimulation of MLR → RVLM neurons on arterial pressure (AP) and heart rate (HR). In urethane-anesthetized rats, in which the MLR neurons expressed ChIEF-tdTomato or palGFP (control) via AAV injections to the MLR (Fig. 1e–h), the RVLM was bilaterally illuminated using 5-ms-pulsed blue laser light (473 nm wavelength) with 10 mW output (when continuously activated) to photostimulate the MLR → RVLM neuronal axons through optical fibers inserted into the brain (Supplementary Fig. 2a). A pulse series at either 20 or 40 Hz for 2 min consistently elicited pressor and tachycardiac responses in ChIEF-tdTomato-expressing rats but not in palGFP-expressing controls (Supplementary Fig. 2b, c). These observations indicate that excited MLR → RVLM neurons elicit autonomic cardiovascular responses, which prompted us to directly examine the sympathoexcitatory role of the MLR → RVLM

monosynaptic pathway with electrophysiological recording of renal sympathetic nerve activity (RSNA).

In anesthetized rats, in which RVLM-projecting neurons expressed channelrhodopsin-2 (ChR2) tagged with GFP or control EGFP via bilateral injections of AAVrg into the RVLM (Supplementary Fig. 1a, b), the MLR was bilaterally illuminated to photostimulate the cell bodies/dendrites of MLR → RVLM neurons (somata targeting) in an intermittent manner, i.e., via a 0.5-s pulse series (5-ms-pulsed blue laser light at 10, 20, or 40 Hz, 10-mW laser output) with a 1.5-s interval for 1 min; the effects of this procedure on RSNA were then examined (Fig. 2a). The photostimulation elicited renal sympathoexcitation (RSNA) that was synchronous with each bout of 0.5-s illumination pulses and was accompanied by an increase in AP but not HR throughout the 1-min stimulation period (Fig. 2b). Superimposing and averaging analyses on RSNA changes over 30 bouts of interventions (Fig. 2c) showed that the pulse series at 10 or 20 Hz but not 40 Hz significantly elicited renal sympathoexcitation, which was followed by rapid sympathoinhibition and a return to pre-photostimulation levels (Fig. 2d). The sympathoexcitatory component in response to photostimulation at 40 Hz, as assessed by the area under the curve (AUC) for the changes in RSNA (Fig. 2c), was 29% smaller than that at 20 Hz ($P = 0.034$, two-sided paired $t$-test), perhaps due to anesthetic effects since photostimulation at 40 Hz in nonanesthetized decerebrated rats significantly increased RSNA as that at 20 Hz (described later; Fig. 3). Urethane anesthesia might amplify the effect of higher frequency photostimulation-recruited recurrent inhibition within the sympathoregulatory circuitry. Indicating the specificity of the sympathoexcitatory/sympathoinhibitory responses elicited by optogenetic stimulation of MLR → RVLM neurons, no RSNA change was elicited by illumination of EGFP-expressing, RVLM-projecting MLR neurons in control rats (Supplementary Fig. 3a).

Because MLR → RVLM neurons express VGLUT2 in their axonal terminals (Fig. 1h) and many RVLM neurons express ionotropic glutamate receptors[34], we examined whether glutamatergic transmission in the RVLM contributes to MLR → RVLM neuron-mediated sympathoexcitation. The RSNA response to 1-min intermittent optogenetic stimulation (at 20 Hz) of MLR → RVLM neurons was tested 10–20 min after bilateral injections into the RVLM with saline (50 nL) or a mixture of 2-amino-5-phosphonovaleric acid and cyanquixaline (AP5/CNQX; 10 mM in saline; 50 nL). Compared with saline treatment, AP5/CNQX treatment led to a 23% decrease of the optogenetically elicited RSNA response, as assessed by AUC values for the RSNA changes (Fig. 2e; Supplementary Fig. 3b). At 60 min after AP5/CNQX treatment, the RSNA response recovered to 90% of that after saline treatments ($P = 0.58$; two-sided paired $t$-test). Taken together, these results indicate that glutamatergic transmission in the RVLM contributes to the MLR → RVLM signaling that drives renal sympathoexcitation.

### Excited MLR → RVLM neurons concomitantly activate sympathetic and somatomotor efferents

Given that glutamatergic MLR neurons play an important role in evoking locomotion[33,35–37], we questioned whether excitation of MLR → RVLM neurons elicits not only sympathoexcitation but also motoneuron excitation, namely central command activation. To investigate this possibility, we simultaneously recorded discharges in peripheral sympathetic and somatomotor efferent nerve fibers in decerebrated, nonanesthetized, and paralyzed rats. In this preparation, whereas the loss of inhibition from corticothalamic circuits leads to overactivity of the brainstem, it is advantageous that the effects of anesthesia and movement feedback can be discarded. The decerebrated rats, in which RVLM-projecting neurons expressed ChR2-GFP, received laser pulses to the MLR unilaterally in the intermittent manner for 1 min as stated above (20 mW laser output; Fig. 3a). Optogenetic stimulation of MLR → RVLM neurons elevated AP without significant effects on HR, and increased discharges in both renal

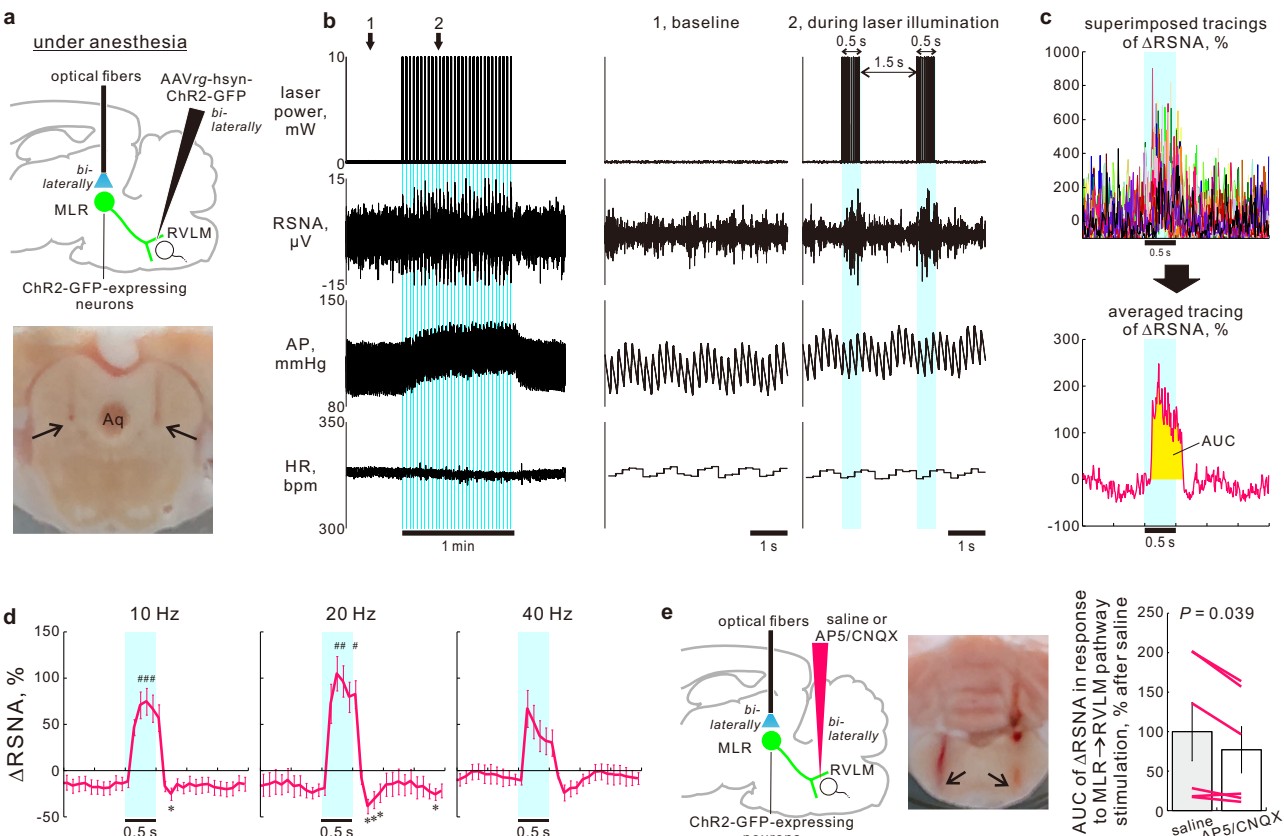

**Fig. 2 | MLR → RVLM pathway drives sympathoexcitation. a** Optogenetic stimulation of MLR → RVLM neurons in anesthetized rats. Arrows: locations of optic-fiber tips. **b** Representative recording during optogenetic stimulation (20 Hz) in a ChR2-GFP-expressing anesthetized rat. Aqua blue-colored backgrounds indicate illumination periods. **c** Superimposing and averaging analysis performed on RSNA shown in panel **b**. Time courses (10-ms bins) of percentage changes in RSNA from the baseline (= 30-s averaged value before the first optogenetic intervention) during each cycle (upper) were averaged at each time point over 30 interventions (lower). **d** Time courses (100-ms bins) of ΔRSNA, after superimposing and averaging analysis, in response to intermittent MLR → RVLM neuron stimulation in ChR2-GFP-expressing anesthetized rats [$n$ = 9 (7 males, 2 females) besides $n$ = 7 (5 males, 2 females) at 10 Hz]. **e** Effect of glutamate receptor blockades in the RVLM on RSNA response to MLR → RVLM neuron stimulation (20 Hz) in ChR2-GFP-

expressing anesthetized male rats ($n$ = 6). Arrows: locations of pipette tips for RVLM injections. Data were analyzed by Friedman one-way RM ANOVA by rank (as the normality or equal variance test was not passed) followed by Dunnett's post hoc test (**d**) or by a two-sided paired $t$-test (**e**). Statistic information including F/t/ χ2 statistic values and degrees of freedom is presented in Supplementary Table 15. *$P$ < 0.05 vs. 30-s averaged baseline. #$P$ < 0.05 vs. 1-s averaged values immediately before each photostimulation over 30 interventions. Baseline values for panels **d** and **e** are reported in Supplementary Tables 2 and 4, respectively. Data shown are means ± SEM. Source data are provided as a Source Data file. The brain section images used in figures (**a**, **e**) were adapted from "Paxinos G & Watson C. The rat brain in stereotaxic coordinates. 6th edn. (Amsterdam, Academic Press/Elsevier, 2007)".

sympathetic and somatomotor L5 ventral root[38] nerve fibers (Fig. 3b). Notably, the discharging manner of these nerve fibers during optogenetic interventions differed; renal sympathoexcitation (RSNA) was elicited immediately and synchronously with each bout of 0.5-s illumination pulses, whereas ventral root nerve activity (VRNA) was gradually elevated and sustained throughout 1-min interventions (Fig. 3b–d). The sustained increases in VRNA are consistent with those in previous studies, which examined the effect of electrical stimulation of the MLR in decerebrate rats[21] and cats[39]. Such differences in the characteristics of sympathetic and somatomotor neuronal discharge responses suggest that medullospinal circuits between MLR → RVLM neurons and spinal motor neurons play a role as a lower-pass filter, whereas distinct medullospinal circuits downstream of the MLR → RVLM pathway mediate the rapid sympathetic nerve responses. In contrast to the ChR2-mediated activation of sympathetic and somatic motor outflows, illumination in EGFP-expressing controls elicited no changes in VRNA (Supplementary Fig. 3c, d).

Using decerebrate rats, we also tested whether the motoneuron excitation and cardiovascular responses elicited by stimulation of MLR → RVLM neurons involves glutamatergic neurotransmission in the RVLM. Bilateral AP5/CNQX injections in the RVLM significantly

reduced activation of VRNA and elevation of AP in response to 15-s sustained photostimulation of MLR → RVLM neurons by 64 and 34% (assessed via integration of VRNA and AP changes during the 15-s stimulation period), respectively, compared to those after saline injections (Fig. 3e; Supplementary Fig. 3e). The VRNA and AP responses recovered 60 min after AP5/CNQX treatment by 105 and 81% of the responses after saline treatment ($P$ = 0.62 and 0.16; two-sided paired $t$-test; $n$ = 3), respectively. In contrast, optogenetic stimulation had no effect on HR (Supplementary Fig. 3e). Altogether, excited MLR → RVLM neurons concomitantly increase sympathetic vasoconstrictor and somatomotor tones partly via glutamatergic transmission in the RVLM.

## Excited MLR → RVLM neurons elicit locomotion and cardiovascular responses

The MLR → RVLM pathway-driven activation of both sympathetic and somatomotor nervous systems led us to investigate whether stimulation of this pathway elicits both locomotor activities and autonomic cardiovascular changes as seen in running exercise. Rats in which RVLM-projecting neurons expressed ChR2-GFP or control EGFP via bilateral AAVrg injections into the RVLM underwent MLR illumination through optical fibers and their AP was monitored with a pressure

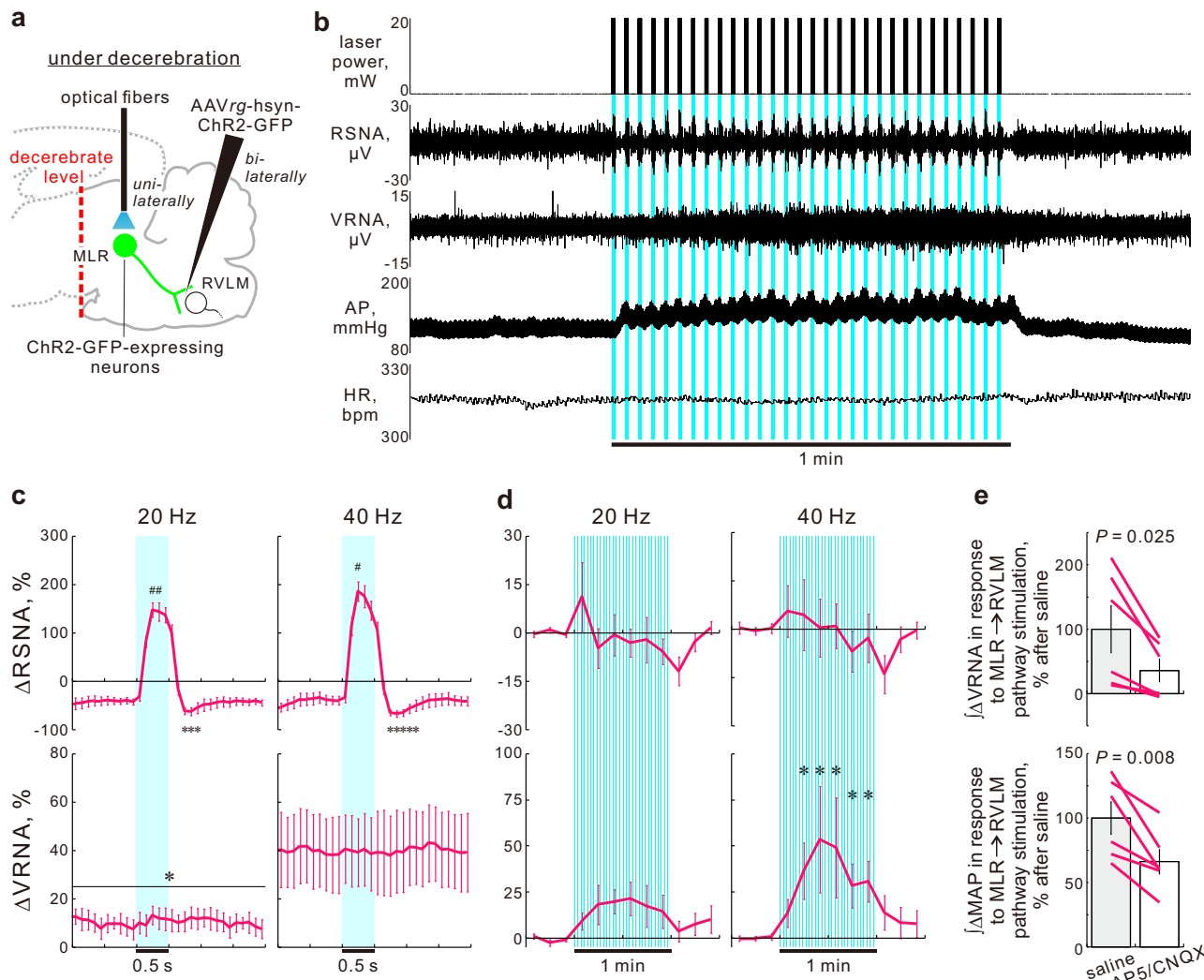

**Fig. 3 | MLR → RVLM pathway drives both sympathoexcitation and motor neuroexcitation. a** Optogenetic stimulation of MLR → RVLM neurons in unanesthetized decerebrated rats. **b** A representative recording during optogenetic stimulation (40 Hz) in a ChR2-GFP-expressing decerebrated rat. **c, d** Time courses after superimposing and averaging analysis (**c** 100-ms bins) and throughout optogenetic interventions (**d** 10-s bins) of ΔRSNA and ΔVRNA in response to 1-min intermittent stimulation of the MLR → RVLM neurons in ChR2-expressing decerebrated male rats (*n* = 6). **e** Effect of glutamate receptor blockades in the RVLM on ΔVRNA and ΔAP responses to 15-s sustained stimulation of MLR → RVLM neurons (40 Hz), assessed by their integrated values for 15 s, in ChR2-GFP-expressing,

decerebrated male rats (*n* = 6). Data were analyzed by one-way RM ANOVA/Friedman one-way RM ANOVA by rank followed by Dunnett's post hoc test (**c, d**) or by a two-sided paired *t*-test (**e**). Statistic information including F/t/χ² statistic values and degrees of freedom is presented in Supplementary Table 15. *$P < 0.05$ vs. 30-s averaged baseline. #$P < 0.05$ vs. 1-s averaged values immediately before each photostimulation over 30 interventions. Baseline values for panels (**c, d**) and (**e**) are reported in Supplementary Tables 5 and 7, respectively. Data shown are means ± SEM. Source data are provided as a Source Data file. The brain section image used in figure (**a**) was adapted from "Paxinos G & Watson C. The rat brain in stereotaxic coordinates. 6th edn. (Amsterdam, Academic Press/Elsevier, 2007)".

telemeter under free-moving conditions. Conscious rats were placed in a circular track (I.D.100 & O.D.140 cm), and when they were resting but not sleeping or moving (e.g., grooming), the cell bodies of MLR → RVLM neurons were illuminated under continuous recordings of AP, HR and behaviors (Fig. 4a–c).

Unilateral illumination of ChR2-GFP-expressing, MLR → RVLM neuron cell bodies for 15 s with pulsed blue laser light at 40 Hz (20 mW laser output) immediately increased AP and belatedly elicited tachycardia. Throughout the stimulation period, the photostimulation also elicited full-body locomotion or running without impeding the ability to brake or turn (Fig. 4d, e; Supplementary Fig. 4a; Supplementary Movie 1). Moreover, five-times-repeated unilateral optogenetic interventions in which one cycle consisted of 5-s laser-on and 5-s laser-off at 40 Hz (20 mW) also elicited pressor responses and subsequent tachycardiac and locomotor responses, which were faithfully synchronous with each bout of 5-s illumination

pulses (Fig. 4f, g; Supplementary Fig. 4c; Supplementary Movie 2). The distances the rats walked/ran and cardiovascular responses during these photostimulations (assessed by integration during the stimulation period) were frequency dependent among 10, 20, and 40 Hz (at 20 mW) and intensity dependent among 10, 20, and 35–40 mW (at 40 Hz) (Supplementary Fig. 4b, d). Bilateral optogenetic stimulation elicited locomotor and cardiovascular responses similarly to those caused by unilateral stimulation (Fig. 4e, g). However, MLR illumination in EGFP-expressing controls did not evoke locomotion or cardiovascular changes (Supplementary Fig. 5). These results demonstrate that excited MLR → RVLM neurons drive both locomotion and cardiovascular responses, which supports the notion that central command signals transmitted through the MLR → RVLM pathway play a role in the coordination of locomotor limb movements and autonomic cardiovascular controls required for running exercise.

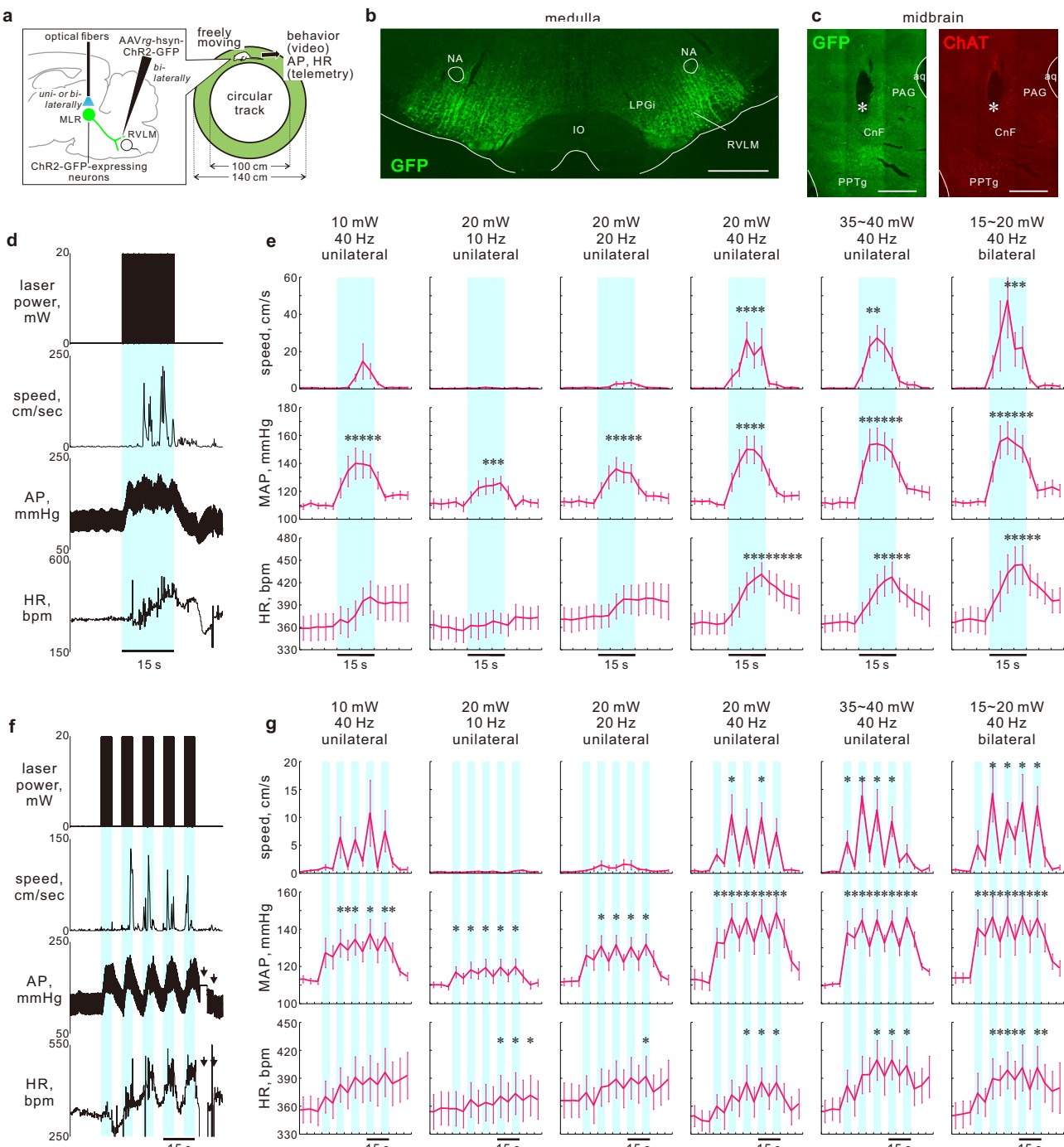

**Fig. 4 | Locomotion and cardiovascular responses driven by the MLR → RVLM pathway. a** Optogenetic stimulation of MLR → RVLM neurons in conscious rats left free to move on a circular track. **b** GFP transduction in the axonal fibers of the medulla of a ChR2-GFP-expressing rat (autofluorescence). Scale bars: 1 mm. **c** GFP and ChAT-immunoreactive cells in the midbrain. Asterisks: a location of the implanted optic-fiber tip. Scale bars: 1 mm. **d** Representative recording during 15-s sustained optogenetic intervention (40 Hz; 20 mW) in a ChR2-GFP-expressing conscious rat. **e** Time course (3-s bins) changes in response to 15-s sustained optogenetic stimulation (*n* = 8, besides *n* = 7 for bilateral stimulation, all males). Tracings of each rat are presented in Supplementary Fig. 4a. **f** Representative recording during five-time-repeated intermittent (5-s laser-on/5-s laser-off) optogenetic interventions (40 Hz; 20 mW) in a ChR2-GFP-expressing conscious rat. Arrows: disconnection of telemetry signals. This recording was included in the

results because the disconnection occurred after optogenetic interventions. **g** Time course (5-s bins) changes during five-times-repeated optogenetic stimulations (*n* = 8, besides *n* = 7 for bilateral stimulation). Tracings of each rat are presented in Supplementary Fig. 4c. Data were analyzed using one-way RM ANOVA/ Friedman one-way RM ANOVA by rank followed by Dunnett's post hoc test [**e**, **g**]. Statistic information including F/t/χ² statistic values and degrees of freedom is presented in Supplementary Table 15. *$P < 0.05$ vs. 15-s averaged baseline. The overhead views during recordings shown in panels (**d**) and (**f**) are reported in Supplementary Movies 1 and 2, respectively. Baseline values for panels (**e**) and (**g**) are reported in Supplementary Tables 8 and 9, respectively. Data shown are mean ± SEM. Source data are provided as a Source Data file. The brain section image used in figure (**a**) was adapted from "Paxinos G & Watson C. The rat brain in stereotaxic coordinates. 6th edn. (Amsterdam, Academic Press/Elsevier, 2007)".

Optogenetic stimulation of MLR → RVLM neuronal cell bodies did not elicit tachycardia in anesthetized or decerebrate rats (Figs. 2, 3) probably because the RVLM predominantly controls sympathetic vasomotor activity rather than cardiac functions[25-28]. Meanwhile, this stimulation in conscious rats increased HR (Fig. 4). The tachycardiac response might be secondary to the locomotor response. Activation of the skeletal muscle-based reflex (i.e., exercise pressor reflex) due to the movement or limb muscle contraction in conscious rats should be involved in the HR response[40]. It is also likely that MLR → RVLM pathway stimulation subsequently activates other central circuits that in turn elicit cardiovascular responses. For example, the MLR regulates cortical state in parallel with locomotion[41].

We also examined whether another RVLM-projecting, sympathoexcitatory pathway mediates locomotion. The hypothalamic paraventricular nucleus (PVN) contains an abundance of RVLM-projecting neurons, of which selective stimulation in anesthetized rats elicits sympathoexcitation[42]. We found that optogenetic stimulation of the PVN → RVLM pathway in freely moving and conscious rats increased AP but never elicited locomotion (Supplementary Fig. 6). Thus, unlike the MLR → RVLM pathway, the sympathoexcitatory PVN → RVLM pathway is not involved in somatic motor control for locomotion.

### Inhibition of MLR → RVLM neuronal excitation during spontaneous running exercise attenuates locomotion and cardiovascular responses

To examine the necessity of MLR → RVLM neurons in somatic-autonomic motor integration for locomotor activities, we tested whether the inhibition of this pathway during voluntary running exercise attenuates both locomotor activity and cardiovascular responses. To optogenetically inhibit MLR → RVLM neurons, we combined the Cre-dependent expression system with anterograde and retrograde AAVs to selectively transduce MLR → RVLM neurons with a chloride-conducting channelrhodpsin iChloC fused with mCherry or control eYFP; the MLR was then bilaterally illuminated with blue laser pulses[43] (Fig. 5a). iChloC-mCherry or eYFP was expressed in 79 ± 5% of Cre-expressing MLR neurons ($n = 10$ rats, in which Cre immunostaining was successful; Fig. 5b). The iChloC-mCherry/eYFP-labeled axons were abundantly distributed in the RVLM, in which the axonal terminals were opposed to RVLM C1 neurons (Fig. 5c), suggesting synaptic contacts between the labeled MLR → RVLM neurons and the RVLM C1 neurons.

Conscious rats that were equipped for optogenetic manipulation and telemetric AP measurements were allowed to move freely in a cube-shaped cage [45 × 45 × 40 (height) cm] containing a horizontal running wheel with a 36-cm diameter (Fig. 5a). During spontaneous running on the wheel or at rest on the cage floor, the MLR was bilaterally illuminated for 2 s with 50-ms pulsed laser (10 mW at 10 Hz)[43]. This optogenetic approach was confirmed to effectively inhibit MLR → RVLM neuronal excitation by immunohistochemical staining of Fos. Fos expression was induced in MLR → RVLM neurons by ChR2-mediated optogenetic activation of their cell bodies under anesthesia, and this induced Fos expression was suppressed by simultaneous photoactivation of iChloC in these neurons (Supplementary Fig. 7).

In each rat with iChloC-mCherry- or control eYFP- expression in MLR → RVLM neurons, experiments were performed over 2–4 days, in which the laser pulse series was totally given 9–17 trials during running and 5–15 trials at rest. Optogenetic interventions during voluntary wheel running were followed by various patterns of behavioral changes, such as re-running after stopping/slowing running or standing on the wheel after stopping running, among trials in each rat (Supplementary Fig. 8a; Supplementary Movie 3). However, the probability that we encountered the behavior of "running through" the 5-sec period after the onset of 2-s laser pulse series was 79% (50–100%, $n = 4$) for eYFP-expressing controls but only 14% (0–33%, $n = 7$) for iChloC-

mCherry-expressing rats (Supplementary Fig 8b). In contrast, the probabilities that locomotor activity was suppressed during the 5-s period were greater for iChloC rats than for controls (Supplementary Fig. 8b); running with a pause or slowing within the 5-s period was seen in 37% (13–57%) of running trials for iChloC rats whereas this behavioral pattern was not observed in any controls. "Standing on the wheel after stopping running" at 5 s after the laser pulse onset was observed in 7% (0–21%) for controls but in 24% (14–35%) for iChloC rats. Moreover, leaving the wheel for the cage floor within the 5-s period was observed in 8–30% of the trials for six of 7 iChloC rats whereas 3 control rats never displayed this behavior besides one control that left the wheel in 29% of the trials.

By averaging changes in wheel rotation rate (WRR) and cardiovascular changes over all running trials for each rat, we found that the optogenetic inhibition of MLR → RVLM neurons during wheel running exerted inhibitory effects on locomotor activity and AP elevation without affecting HR (Fig. 5d–f). Since both WRR and AP reductions occurred immediately after the onset of illumination and exhibited similar time course kinetics on average (Fig. 5e; Supplementary Fig. 8c), the suppressed locomotion was unlikely a cause of the AP reduction or vice versa. In eYFP-expressing controls, meanwhile, the laser pulse series given during running had no effect on WRR or cardiovascular changes on average (Fig. 5e and f, Supplementary Fig. 8c). Finally, in 20% (0–31%) of the trials for controls but in 71% (62–82%) for iChloC rats, concomitant decreases in WRR and AP following optogenetic intervention, irrespective of behavioral patterns, were observed (Fig. 5g and h). These results indicate that MLR → RVLM neurotransmission mediates the central command signaling that is engaged in the coordination of locomotor activity and sympathetic cardiovascular controls during running exercise.

In contrast, laser pulse series given while rats were at rest had no effect on AP or HR in either controls or iChloC-mCherry-expressing rats (Fig. 5i; Supplementary Fig. 9). Thus, the MLR → RVLM monosynaptic pathway is unlikely to contribute to basal cardiovascular homeostasis.

## Discussion

Our functional neuroanatomical analyses and in vivo physiological experiments combined with optogenetic manipulation in rats revealed that MLR neurons transmit central command-driven excitatory, at least partly glutamatergic, signals onto the RVLM that stimulate both somatomotor and sympathetic outflows during voluntary exercise. Optogenetic stimulation of the MLR → RVLM monosynaptic pathway elicited locomotor and autonomic cardiovascular responses as seen in running exercise. Moreover, selective inhibition of the MLR → RVLM pathway suppressed locomotor activity and reduced the pressor response during voluntary wheel running, but it did not affect basal cardiovascular homeostasis under resting conditions. Overall, these findings demonstrate that the subcortical MLR → RVLM pathway constitutes a key component of the central circuit mechanism that relays volitional central command signals and thereby mediates the coordination of autonomic cardiovascular control and somatomotor limb control, which is required to enhance the performance of locomotion or running exercise.

Along with the MLR → RVLM pathway, the RVLM also receives projections from the PAG, as shown in Fig. 1c. Although the PAG appears to constitute another subcortical circuit for central command[19], whether RVLM-projecting PAG neurons are engaged in autonomic regulation during exercise is unknown and deserves further study.

Whereas somatomotor and autonomic control systems have been traditionally considered disparate[44], the primitive behaviors of vertebrates, including exercise, locomotion, feeding, sleeping, the fight-or-flight-or-freeze reaction, and the pain reflex, are characterized by coordinated somatomotor and autonomic regulation[45-47]. Thus,

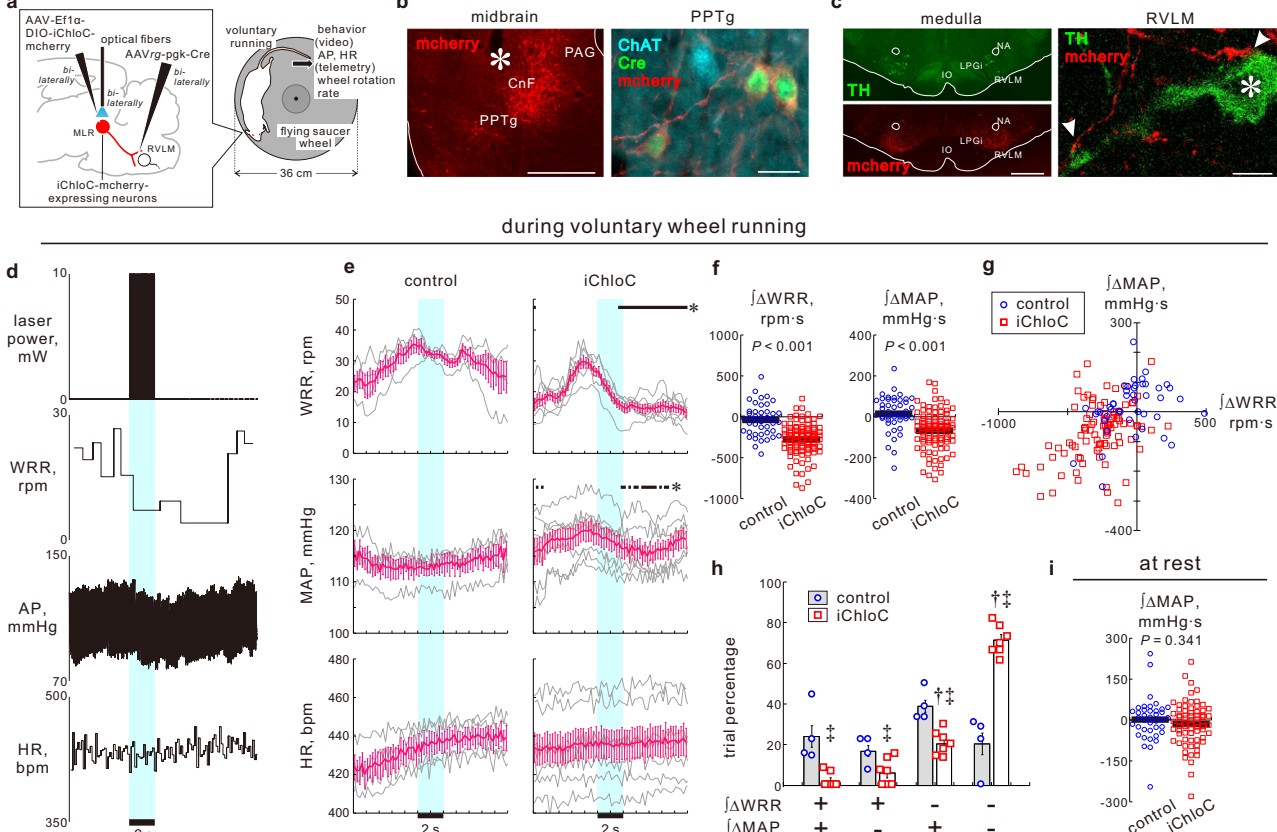

**Fig. 5 | MLR → RVLM pathway mediates both locomotor activities and cardiovascular responses during voluntary running exercise. a** Optogenetic inhibition of MLR → RVLM neurons of conscious rats free to move in a cage containing a flying saucer wheel. **b** Cre-dependent iChloC-mCherry expression in MLR → RVLM neurons. Asterisk: optic-fiber tip location. Scale bars: 1 mm (left); 20 μm (right). **c** Distribution of mCherry-immunoreactive axons in the medulla (left); swellings were closely opposed to RVLM C1 neuronal dendrites and cell bodies (asterisks; right). Scale bars: 1 mm (left); 10 μm (right). **d** Representative locomotor speed and cardiovascular changes in response to 2-s optogenetic inhibition during wheel running in an iChloC-mcherry-expressing rat. WRR, wheel rotation rate. The overhead view during recording is reported in Supplementary Movie 3. **e** Time course (200-ms bins) changes in response to 2-s optogenetic intervention during running in eYFP-expressing controls ($n = 4$) and iChloC-mCherry-expressing male rats ($n = 7$). Gray, individual data averaged over trials. **f** Comparisons between controls (45 trials/4 rats) and iChloC-mCherry-expressing rats (96 trials/7 rats): integrated ΔWRR and ΔMAP for 5 s after the onset of 2-s optogenetic intervention during running. Pre-illumination level was determined as a mean value during the 2-s period immediately prior to laser illumination. Horizontal bars, mean values. **g** Plots

of ∫ΔMAP vs. ∫ΔWRR in each running trial. Many plots of iChloC-mCherry-expressing rats distributed in the third quadrant. **h** Comparisons of trial percentages between controls and iChloC-mCherry-expressing rats and among patterns of ∫ΔWRR and ∫ΔMAP. **i** Comparison between controls (39 trials/4 rats) and iChloC-mCherry-expressing rats (69 trials/7 rats) of ∫ΔMAP at rest. Data were analyzed via one-way RM ANOVA followed by Holm–Sidak's post hoc test or Friedman one-way RM ANOVA by rank followed by Dunnett's post hoc test (**e**), two-sided Welch's *t*-test (**f**), two-way RM ANOVA followed by Tukey's test (**h**), or Mann–Whitney *U* test (**i**). Statistic information including F/t/χ² statistic values and degrees of freedom is presented in Supplementary Table 15. *$P < 0.05$ vs. 2-s averaged pre-illumination level immediately prior to optogenetic intervention (**e**). †$P < 0.05$ vs. all other trials in each group (**h**). ‡$P < 0.05$ vs. controls in each pattern (**h**). Baseline values for panels (**e**–**h**) and (**i**) (= pre-illumination level) are reported in Supplementary Tables 13 and 14, respectively. Data shown are means ± SEM. Source data are provided as a Source Data file. The brain section image used in figure (**a**) was adapted from "Paxinos G & Watson C. The rat brain in stereotaxic coordinates. 6th edn. (Amsterdam, Academic Press/Elsevier, 2007)".

functional brain architecture likely underlies the fine coordination of somatomotor and autonomic activities for specific behaviors, yet its organization is currently poorly understood. In potential relevance to this circuit, double transneuronal tracings in rats revealed the presence of neuronal populations with connections to both somatic and sympathetic motor systems in discrete regions throughout the brain[48–50]. Moreover, electrical or chemical stimulation of an area in the hypothalamus or midbrain simultaneously caused somatomotor and cardiovascular changes in animals and humans[13–15,17,18]. Regions that have previously been neuroanatomically or functionally investigated, such as PVN[50], MLR[15,49], RVLM and lateral paragigantocellular nucleus (LPGi) in the caudal medullary reticular formation[48] among others[13–15,17,18,48–50], may participate in the somatic-autonomic motor integration and thereby affect different behaviors. However, the causal roles of these regions in behaviors as well as brain connectivity underlying these roles have not been explored. In the present study,

we identified the MLR → RVLM pathway as a central mechanism underlying the somatic-autonomic motor integration for voluntary running exercise. Consequently, our study provides a key insight into the brain architecture for physiological conditioning necessary to maximize exercise performance.

The MLR → RVLM pathway appears to constitute a key connection through which volitional central command signals for locomotion or voluntary running exercise affect the somatomotor and sympathetic nervous systems. As evidenced by the different patterns of somatomotor and sympathetic nerve discharges evoked by MLR → RVLM neuron stimulation (Fig. 3b–d), excitatory signals from this pathway seem to drive distinct medullospinal pathways connecting to spinal motoneurons and sympathetic preganglionic neurons. However, the intrinsic organization to functionally link the MLR → RVLM pathway with either locomotor activities or sympathetic cardiovascular responses has been unclear. Nevertheless, our tracing study indicates

that postsynaptic neurons targeted by the MLR → RVLM glutamatergic pathway include RVLM C1 neurons (Figs. 1h and 5c), although RVLM non-C1 neurons may also be a target of this pathway. Both C1 and non-C1 neurons in the RVLM send axonal projections to sympathetic preganglionic neurons in the spinal cord[24]. Central command signals through the MLR → RVLM pathway likely increase sympathetic vasomotor outflows through these sympathetic premotor RVLM neurons.

Regarding the postsynaptic connectivity of MLR → RVLM neurons for somatomotor nervous system regulation, we note that locomotion cannot be initiated by non-selective stimulation of RVLM neurons in conscious rodents[51,52], and this view is also supported by no effect of sympathoexcitaotry PVN → RVLM neuronal stimulation on rat behavior (Supplementary Fig. 6). Thus, the locomotion driven by MLR → RVLM neurons seems to require activation of a specialized subpopulation of RVLM neurons, that is specifically controlled by this monosynaptic pathway and engages executive medullospinal locomotor circuits. Key intermediaries between MLR → RVLM neurons and spinal locomotor neurons may include glutamatergic LPGi neurons. Capelli et al.[35] showed that conditional ablation of this neuronal population suppressed glutamatergic MLR neuron-elicited locomotion in VGLUT2-Cre transgenic mice, suggesting that glutamatergic LPGi neurons play a modulatory role in the positive tuning of locomotion mediated by glutamatergic MLR neurons. The RVLM investigated in our experiments is positioned continuously caudolaterally to the LPGi investigated by Capelli et al. rostrocaudally lying to the caudal edge of the facial nucleus according to morphological data in the previous[35] and our studies (based on the distribution of C1 neurons and the brain areas for AAV transduction/optic cannula implantation), as well as the standard mouse and rat brain atlases. As both the RVLM and LPGi are parts of the medullary reticular formation, which is characterized by the interconnected structure and expansive network of tracts, MLR → RVLM neurons may have synaptic contacts with a specific RVLM neuronal population that regulates glutamatergic LPGi neurons and/or other medullospinal locomotor neurons. It is also possible that locomotor-controlling LPGi neurons are intermingled in the medial part of the RVLM; they may be postsynaptically controlled by MLR → RVLM neurons. Overall, while central command signals descending through MLR → RVLM neurons regulate the sympathetic nervous system via activation of sympathoregulatory RVLM neurons, the signals may collaterally stimulate another RVLM neuronal population, which in turn activates a distinct medullospinal circuitry network including the LPGi for locomotion. Further investigations are required to precisely determine the separate downstream circuit mechanisms from MLR → RVLM neurons for locomotor activities and sympathetic cardiovascular responses.

Locomotion is a representative of various voluntary behaviors, including escape, pursuit, and exploration, which are elicited by different motor volitions. It is uncertain whether the MLR → RVLM pathway, which plays an essential role in voluntary running exercise, is also engaged in other locomotor behaviors. Nevertheless, activation of the MLR certainly contributes to locomotion for escape[53] and pursuit[54]. Caggiano et al.[36] also suggested that glutamatergic CnF neurons which receive projections from the periaqueductal gray support fast locomotion required for escape, whereas glutamatergic PPTg neurons, which receive projections from basal ganglia, may be necessary for slow explorative locomotion. Thus, the MLR → RVLM pathway may constitute a common key node that underlies somatomotor–autonomic coupling for these types of locomotor behavior, even though it may receive afferent inputs from distinct circuits driven by different motor volitions. Although evidence is currently lacking, upstream circuits that influence the activity of MLR → RVLM neurons for voluntary running exercise may include brain regions related to internal motivation, such as the hypothalamic orexinergic system[55] and/or nucleus accumbens[56], both of which have axonal projections to the MLR[33,57]. Additionally, although

stimulation of supplemental motor area, premotor area, or motor cortex elicits muscle twitch coupled to sympathoexcitation in human subjects[16], no information is available for cortical–subcortical connections to transmit central command signals. Moreover, whereas central command is speculated to originate in the telencephalon as "cortical irradiation"[2], there is no consensus on the site that serves as the origin of central command signals[58]. Upstream circuits that relay central command signals to MLR → RVLM neurons deserve further investigation towards brain mechanisms underlying autonomic cardiovascular adjustments during voluntary exercise, an outstanding issue for well over a century[1,2] (Supplementary Fig. 10).

## Methods

### Rats

All procedures were approved by the Animal Care Committee (ref#: 15-Y-40, 18-Y-11, 19-Y-53) and the Gene Recombination Experiment Safety Committee (ref#: 28-034, 31-067, 32-061) of Tottori University. The Sprague Dawley rats (Slc:SD; male and female) from Japan SLC, Inc were used in this study; they were purchased from Shimizu Laboratory Supplier Co, Ltd and bred in our facilities. All rats were maintained in an air-conditioned room at 25 °C with a 12:12-h light:dark cycle. They were housed in standard cages, except for rats for experiments to study the effect of optogenetic inhibition via iChloC activation, which were housed (after weaning) in cages that contained a flying saucer wheel (36-cm diameter; Exotic Nutrition). Food and water were made available *ad libitum*. All in vivo experiments were performed in the air-conditioned laboratory at 25 °C.

### AAV vectors

The AAVs to anterogradely transduce MLR neurons with ChIEF-tdTomato (AAV2/1-CMV-ChIEF-tdTomato) and with palGFP (AAV2/1-CMV-palGFP) under a CMV promoter (the gene cassette encoding ChIEF-tdTomato was donated by R. McQuiston: Addgene#32846) and for Cre-dependent transduction of MLR → RVLM neurons with iChloC-mCherry (AAV2/5-Ef1α-DIO-iChloC-mCherry) (the gene cassette encoding iChloC-mCherry was donated by P. Hegemann: Addgene#85467) have been previously described[43,59,60]. The production and purification of these AAVs followed the modified Gene Transfer Targeting and Therapeutics Core protocol at Salk Institute (http://vectorcore.salk.edu/protocols.php). The required plasmids were transfected into HEK293T cells and AAV was purified from a crude lysate of the cells using OptiPrep (Axis-Shield). Following concentration via a membrane filter (Amicon Ultra-15 NMWL 50 K, Merck), the final titrations were $2.1 \times 10^{11}$ GC/mL (AAV2/1-CMV-ChIEF-tdTomato), $3.5 \times 10^{11}$ GC/mL (AAV2/1-CMV-palGFP), and $3.4 \times 10^{12}$ GC/mL (AAV2/5-Ef1α-DIO-iChloC-mCherry). All other AAVs were obtained from Addgene (AAVrg-hsyn-EGFP, donated by B. Roth: Addgene#50465, $7.4 \times 10^{12}$ GC/mL; AAVrg-Syn-ChR2(H134R)-GFP, donated by E. Boyden: Addgene#58800, $8.0 \times 10^{12}$ GC/mL; AAVrg-pgk-Cre, donated by P. Aebischer: Addgene#24593, $9.5 \times 10^{12}$ GC/mL; AAV; AAV2-Ef1a-DIO-EYFP, donated by K. Deisseroth: Addgene#27056, $3.0 \times 10^{12}$ GC/mL; AAV5-EF1a-DIO-ChR2-eYFP, donated by K. Deisseroth: Addgene#35509, $1.0 \times 10^{12}$ GC/mL). Plasmids and AAVs with the Addgene numbers provided here were obtained under material transfer agreements with Addgene.

### Brain injections and implantations

Rats (>6 weeks old) anesthetized with 1–5% isoflurane in oxygen, intubated, and artificially ventilated (SN480-7, Shinano) were positioned in a stereotaxic head unit (900LOS from David Kopf Instruments, Inc., or SR-6R from Narishige). CTb or given AAV solution was injected into the brain site of interest by using a calibrated pressure-microinjection system (Nanoject II, Drummond Scientific Co.). For injections to the RVLM, the dorsal surface of the medulla was exposed by a midline incision made through the skin covering the back of the

head, followed by dissection of the muscles overlaying the base of the skull, and then an incision made through the atlanto-occipital membrane. The coordinates for the RVLM injections (1.0 mm rostral and 1.8 mm lateral to the calamus scriptorius; 3.5–3.7 mm ventral to the dorsal surface of the medulla) corresponded to those located approximately caudally to the caudal pole of the facial nuclei. For injections to the MLR (8.0 mm caudal, 2.0 mm lateral, and 6.3–6.9 mm ventral to the bregma), the skull was exposed by a midline incision of the skin and two burr holes were made in the skull. The solutions injected into the brain were as follows: Alexa-555-conjugated CTb (1.0 mg/1 mL PBS, C34776, Thermo Fisher Scientific) (23.0 nL × 4, RVLM), AAV-CMV-ChIEF-tdTomato (46.0 nL × 4, MLR), AAV-CMV-palGFP (46.0 nL × 4, MLR), AAV-Ef1α-DIO-iChloC-mCherry (46.0 nL × 4, MLR), AAV- Ef1α-DIO-EYFP (46.0 nL × 4, MLR), AAV-Ef1α-DIO-ChR2-eYFP (46.0 nL × 4, MLR), a mixture of AAV-Ef1α-DIO-ChR2-eYFP and AAV-Ef1α-DIO-iChloC-mCherry (1:1, 46.0 nL × 4, MLR), AAVrg-hsyn-EGFP (23.0 nL × 3, RVLM), AAVrg-Syn-ChR2(H134R)-GFP (23.0 nL x 3, RVLM), and AAVrg-pgk-Cre (23.0 nL × 3, RVLM). After injections, the micropipette remained inserted for 5 min before it was withdrawn.

In CTb-injected rats, a recovery period of 7–11 days was allowed until they were subjected to transcardial perfusion. AAV-injected rats used in experiments to examine Fos expression in RVLM-projecting neurons were allowed a recovery period of 7 days before the treadmill exercise training program began. In the AAV-injected rats used for optogenetic experiments under anesthetized or decerebrated states, a period of at least 14 days was allowed before the experiments. AAV injection procedures for rats in conscious experiments followed by an additional surgery to intracranially implant fiber-optic cannulas (200-μm core diameter, CFML52U-20, Thorlab). Two optical fibers for bilateral illumination of the MLR were vertically inserted to the brain (8.0 mm caudal, 2.0 mm lateral, and 5.5–5.8 mm ventral to the bregma) through two burr holes made in the skull, and secured along with the screws placed surrounding the holes using dental cement. For unilateral illumination of the PVN, a fiber was likewise vertically inserted (1.9 mm caudal, 0.3 mm lateral, and 7.9–8.2 mm ventral to the bregma) and secured. The rats used in conscious experiments were allowed more than 7 days before being subjected to another surgery to implant a wireless pressure telemeter for AP measurements (TRM54P, Millar, Inc./Kaha Sciences), which is described below.

In isoflurane-anesthetized, intubated, and mechanically ventilated rats, a midline abdominal incision was made to expose the peritoneal cavity and then a blunt dissection was made to reach the descending aorta. During temporal occlusion of the aorta with a 2-0 silk at the iliac bifurcation and below the left renal artery bifurcation, the pressure sensor of the telemeter was then inserted in the aorta, advanced 1–2 cm rostrally, and fixed with surgical mesh and biocompatible surgical glue. After the aorta occlusion was released, the skin incision was suture-closed. Experiments in conscious rats were conducted at least 1 week after the telemeter implantation.

## Treadmill exercise training and protocol

To immunohistochemically study the excitabilities of the MLR → RVLM pathway by voluntary running exercise, male rats, that had received bilateral injections to the RVLM with AAVrg-hsyn-EGFP, were treadmill exercise-trained 3–4 times per week and for a total of 14–18 days. On the first day of training, the rats were placed on a custom-built treadmill with an electrical shock grid installed at the rear (MK-680C; Muromachi) for at least 30 min. They were then subjected to treadmill running exercise at 10 m/min and a 0˚ incline for 1 min, which was immediately followed a 30-s period of buzzer sound (1 Hz) as a trigger for the exercise onset. This session was repeated 3–5 times for the rats to learn the association between the buzzer sound and exercise onset. After the second training day, the rats had been subjected to one running session per day. The speed and duration for running were gradually increased over 10 days until 20 m/min and 40 min,

respectively. After completing the treadmill exercise at 20 m/min for 40 min once, the rats were subjected to treadmill running exercise at 16 m/min for 40 min for the remaining training sessions. If the running pace dropped below the treadmill rate during each training session, a mild but aversive electrical stimulation of the foot would be provided to rats with the shock grid. However, the rats were administered very few shocks because they received a gentle nudge with a cotton swab when they were about to touch the grid. Consequently, these rats completed the training program and became capable of voluntarily treadmill running at 16 m/min for 40 min without receiving any shocks or nudges. A period of two days without training was allowed between the final training day and the protocol day.

On the protocol day, the rats were randomly chosen as "Exercise" and "Control" group. The Exercise rats were brought to the treadmill, on which foot shocks had never been administered, and maintained a 15-min resting period. Subsequently, after the buzzer sound period for 30 s, they were subjected to 40-min treadmill exercise at 16 m/min. Nudges were unnecessary during this treadmill exercise since the rats were running throughout the period without approaching the rear of the treadmill. The non-exercised Control rats were brought to the treadmill but not subjected to exercise. Ninety minutes after the offset of exercise or the control period, the rats were deeply anesthetized via inhalation of 5% isoflurane in oxygen and then immediately transcardially perfused (as described later).

## In vivo optogenetic experiments in anesthetized/unanesthetized decerebrate rats

For surgeries to measure peripheral nerve activities and cardiovascular parameters, the rat was anesthetized with 1–5% isoflurane in oxygen, intubated, and mechanically ventilated. The right femoral artery and vein were catheterized to measure AP via a pressure transducer (P23XL, Becton, Dickinson & Co.) and to infuse drugs, respectively. Two needle electrodes were placed on the forelimbs to record ECG from which HR was calculated using the time between successive R waves. The rat was placed in the stereotaxic head unit. To measure RSNA, the left kidney was retroperitoneally exposed through a left flank incision and then a bundle of the renal nerve fibers was connected to a bipolar electrode made of stainless-steel wire. To measure L5 VRNA, a laminectomy to expose the lower lumbar portion of the spinal cord and an incision of the meningeal layers surrounding the cord were performed. The nerve bundle obtained from left L5 ventral root was carefully isolated and placed on an insulated bipolar stainless electrode and then the bundle distal to the electrode was cut and ligated. Discharges of the L5 ventral root reflect activities of motoneurons directed to hindlimb skeletal muscle[61] since the ventral roots caudal to the L3 in rats do not carry axons of sympathetic preganglionic neurons[38]. The RSNA and VRNA signals were amplified using an AC amplifier (MEG-5200; Nihon-Kohden) with a bandpass low-frequency filter of 150 Hz and a high-frequency filter of 1 kHz so that they were audible. The exposed neural tissues were immersed in mineral oil.

The rats for experiments under anesthesia received intravenous administration of urethane (600 mg/kg) and α-chloralose (60 mg/kg). In rats for experiments under unanesthetized decerebrated conditions, bilateral carotid arteries were ligated to minimize cerebral hemorrhage during the decerebration procedure. After a parietal craniotomy, the brain was sectioned coronally with a blade at the precollicular level. All neural tissue rostral to the section and the cortical tissues covering the cerebellum were aspirated.

To deliver laser light to the targeted region, fiber-optic cannulas, which were connected to a 473-nm-wavelength diode-pumped solid-state laser (BL473T8-200; Shanghai Laser and Optic, Co.) controlled by a pulse generator (SEN-7103; Nihon Kohden or STOmk-2; BRC Co.) through a mono or branching fiber-optic patch cord were inserted in the brain. Prior to the insertion, the laser output intensity at the tip of

each cannula was measured with a meter (LPM-100; BRC Co.). For bilateral RVLM illumination, two cannulas were inserted into the brainstem at an angle vertical to the dorsal surface of the brainstem (1.0 mm rostral and 1.8 mm lateral to the calamus scriptorius) with cannula tips located at 3.0–3.2 mm rostroventral from the surface. For illumination of the MLR, the tips of the cannulas were inserted perpendicularly into the brain (8.0 mm caudal, 2.0 mm lateral, and 5.5–5.8 mm ventral to the bregma in anesthetized non-decerebrated rats; 0.2–0.5 mm rostral and 2.0 mm lateral to the border of the inferior and superior colliculi, and at 3.5 mm ventral to the dorsal surface of the colliculi in decerebrate rats). After all surgical procedures were completed, the rat was removed from isoflurane. At least 60 min was allowed to pass before the experimental protocols began.

In spontaneously breathing urethane-anesthetized rats, that had received bilateral injections to the MLR with AAV-CMV-ChIEF-tdTomato or AAV-CMV-palGFP, the RVLM was bilaterally laser-illuminated at 20 or 40 Hz with a 5-ms pulses for 30 s. The laser power was preset at 10 mW when continuously illuminated. In urethane-anesthetized or decerebrated rats that had received bilateral injections to the RVLM of AAVrg-Syn-ChR2(H134R)-GFP or AAVrg-hsyn-EGFP, 1-min intermittent (0.5-s pulse illumination with a 1.5-s interval; 30 bouts) or 15-s sustained photostimulation of the MLR at 10, 20, or 40 Hz with a 5-ms laser pulse (10-mW laser power output in anesthetized rats and 20-mW in decerebrate rats) was given. The decerebrated rats were preliminarily paralyzed by intravenous infusion of pancuronium bromide (0.5 mg/kg of body weight). During data collection, the frequency for the mechanical ventilation was set at 70 breaths/min, which was not synchronized with that for the periodic photostimulation (0.5-s laser-on and 1.5-s laser-off). This was conducted to randomize the possible effect of lung inflation-entrained sympathetic outflow on RSNA responses to the periodic photostimulation[21,61]. The order of stimulation frequency was random and intervals of at least 10 min were allowed between maneuvers.

A mixture of ionotropic glutamate receptor antagonists 2-amino-5-phosphonopentanoic acid (AP5; A5282; Merck) and water-soluble 6-cyano-7-nitro-quinoxaline-2,3-dione (CNQX; ab120044; Abcam) (10 mM each in saline) was bilaterally injected to the RVLM to inhibit glutamatergic transmission in the RVLM. Injections were made with the NANOJECT II microinjection system transcranially via burr holes made in the skull (12.5 mm caudal, 1.8 mm lateral, and 10.5–10.8 mm ventral to the bregma) or through the dorsal surface of the medulla as stated above. After the bilateral injections of saline or AP5/CNQX (46.0 nl/site), the duration to photostimulation was 5–20 min.

After data collection was completed, the fiber-optic cannulas and/or micropipettes were repeatedly inserted into and removed from the brain to make scars for post hoc confirmation of the tip locations. The renal nerve and ventral root bundle were cut between the electrode and the neural axis to measure the background noise of RSNA and VRNA, respectively. Rats were deeply anesthetized with an additional intravenous infusion of urethane and α-chloralose or inhalation of 5% isoflurane in oxygen, after which they underwent transcardial perfusion as described later.

**In vivo optogenetic stimulation experiments in conscious rats**
Male rats that had received bilateral injections to the RVLM with AAVrg-hsyn-EGFP or AAVrg-hsyn-ChR2-GFP and been implanted with fiber-optic cannulas as well as a telemetry transmitter were subjected to experiments to observe changes in AP, HR (calculated from AP waves), and behavior in a conscious state when the MLR or PVN was laser-illuminated via the cannulas. Prior to the experimental day, the rats were allowed to move freely for 1–1.5 h on a circular track and become accustomed to the experimental environment. The inner and outer diameters of the track were 100 and 140 cm, whereas the inner and outer wall heights were 30 and 40 cm, respectively. The inner wall was made of black polyvinyl chloride-wrapped polyethylene, whereas the outer wall was made of transparent acrylic board. The ground of the track was made from smooth-surface, dark green or black rubber sheets.

On the experimental day, the fiber-optic cannulas for MLR illumination were connected to a 473-nm-wavelength diode-pumped solid-state laser via a 1.5-m mono or branching fiber-optic patch cord, 1 × 1 fiber-optic rotary joint and FC-FC patch cord. The laser power output for MLR illumination was preliminarily adjusted with other fiber-optic cannulas which had the same laser transmission efficiency as that of the implanted cannulas. The conscious rat was left to move free in the circular track and at least 30 min were allowed before the experiments began. In the resting rat (not sleeping or actively moving such as walking or grooming), the MLR was unilaterally or bilaterally illuminated with 5-ms laser pulses at 10, 20, or 40 Hz after collecting baseline data for 15 s. Light pulses were given sustainedly for 15 s, or intermittently in a five-time repetitive manner for 5-s laser-on with a 5-s interval. In other rats used to study the effect of PVN → RVLM neuronal excitation, the PVN was likewise laser-illuminated unilaterally. In any cases in which disconnection of telemetry signals occurred during optogenetic interventions for more than 1 s, the data were discarded. In cases of disconnection for > 1 s after optogenetic interventions (e.g., Fig. 4f), the data were included in the analyses. In each rat on an experimental day, 2–6 trials were conducted in random order, and intervals of at least 10 min were allowed between trials. Experimental days were at least 2 days apart.

**Fos screening to test the efficacy of optogenetic inhibition via iChloC in vivo**
Isoflurane-anesthetized male or female rats that had received bilateral injections to the RVLM with AAVrg-hsyn-EGFP and to the MLR with AAV-Ef1a-DIO-eYFP, AAV-Ef1a-DIO-ChR2-eYFP, or a mixture of AAV-Ef1a-DIO-ChR2-eYFP and AAV-Ef1a-DIO-iChloC-mcherry were mechanically ventilated and cannulated via a femoral vein. The optical fiber tips were inserted to the brain for MLR illumination (4.5 mm ventral to bregma). Subsequently, under anesthesia with intravenous infusion of urethane and α-chloralose, the MLR was bilaterally and intermittently (10-s on/10-s off, 90 times) illuminated for 30 min with 50-ms-pulsed blue laser (10 mW, 10 Hz). Forty-five min after the offset of illumination, the animals were transcardially perfused with 4% paraformaldehyde, and then the brain was taken for immunohistochemical staining to label Fos expression as described below.

**In vivo optogenetic inhibition experiments in conscious rats**
Juvenile male rats (4–6 weeks old) were group-housed in acrylic glass cages containing a 36-cm diameter-flying saucer wheel (Exotic Nutrition). Consequently, they became willing to spontaneously run on the wheel when they were subjected to the study. After reaching maturity (>9 weeks old), these rats received bilateral injections into the MLR with AAV-Ef1α-DIO-iChloC-mCherry or AAV-Ef1a-DIO-EYFP and into the RVLM with AAVrg-pgk-Cre, and they were implanted with fiber-optic cannulas and a telemetry transmitter (as described above). The experiment was conducted during the dark phase. On the experimental day, the fiber-optic cannulas were connected to the laser via the patch cords and rotary joint, and the power output for MLR illumination was preset at 10 mW.

After the conscious rat was left to roam free in an acrylic glass-made, cube-shape cage [45 × 45 × 40 (height) cm] that contained the flying saucer wheel, a period of at least 30 min was allowed to pass before the experiments began. Four small protrusions were installed in the wheel edge at even intervals; these were faintly contactable to a bar that was placed in the cage and connected to a force transducer (FT03; Grass Instruments) via a spring. Consequently, the wheel rotation rate (WRR) was calculable with time intervals between the tension development events due to the contacts between the protrusions and the bar. While the rat was voluntarily running on the wheel or at rest (not

sleeping or moving, e.g., grooming) on the ground, the MLR was bilaterally illuminated for 2 s with 50-ms pulses at 10 Hz[43]. Illumination during running began at least 2 s after the onset of spontaneous running. In one day, 5–13 trials at rest or during running were randomly conducted for each rat with intervals of at least 5 min allowed between trials. Experimental days for each rat were at least two days apart.

## Perfusion and immunohistochemistry

Rats anesthetized deeply with an additional intravenous infusion of urethane or inhalation of 5% isoflurane in oxygen were transcardially perfused with heparinized saline followed by 4% paraformaldehyde in 0.1-M phosphate-buffered saline (PBS; pH7.4). The brains were removed, post-fixed in the same fixative at 4 °C for 2 h, and then transferred to a 30% sucrose solution at 4°C for 24–48 h. Using a cryostat (CM1900, Leica, Wetzlar, Germany or HM505 E, GMI, Ramsey, MN, USA), 30-μm-thick coronal or sagittal brain sections were produced.

For immunofluorescence staining, the tissue sections were washed in PBS (2 washes × 10 min) and incubated in an antibody solution (PBS containing 0.3% Triton X-100, 2.5 g/L lambda carrageenan, 200 mg/L NaN3, 10 mL/L normal donkey serum) for 2 h at room temperature. The sections were then incubated in a primary antibody solution overnight at 4 °C. The next day, after the sections were rinsed in PBS containing 0.03% Triton X-100 (2 × 10 min), they were incubated in the secondary antibody solution in the dark for 1 h at room temperature, and then rinsed again in PBS (2 × 10 min) in the dark. Finally, the sections were mounted on slides and coverslipped with liquid mountant (P36930; Thermo Fisher Scientific). Digital images of the stained sections were captured with a digital microscope (BZ-9000 or BZ-X710; Keyence) or a confocal microscope (LSM780; Carl Zeiss).

The primary antibodies used were as follows: chicken anti-tyrosine hydroxylase (1:500; ab76442, Abcam), goat anti-choline acetyltransferase (1:100 or 1:200; AB144P, Merck), goat anti-GFP (1:1000; GTX26673, GeneTex, Irvine, CA, USA), goat anti-tdTomato (1:500; AB8181, Sicgen), mouse anti-Cre Recombinase (1:1000; MAB3120, Merck), rabbit anti-c-Fos (1:400; 2250s, Cell Signaling Technology), rabbit anti-GFP (1:1000; A-6455, Invitrogen), rabbit anti-RFP (1:1000; 600-401-379, Rockland), rabbit anti-tyrosine hydroxylase (1:1000; AB152, Merck), and rabbit anti-vesicular glutamate transporter 2 (1:500; AF860, Frontier Institute). The following secondary antibodies used (host species, donkey): anti-chicken DyLight 405 (1:250; 703-475-155, Jackson ImmunoResearch), anti-chicken Alexa Fluor 488 (1:500; 703-545-155, Jackson ImmunoResearch), anti-goat Alexa Fluor 405 (1:500; ab175665, Abcam), anti-goat Alexa Fluor 488 (1:500; ab150129, Abcam), anti-goat Alexa Fluor 555 (1:500; A-21432, Thermo Fisher Scientific or ab150130, Abcam), anti-mouse Alexa Fluor 488 (1:500, ab150109, Abcam), anti-mouse Alexa Fluor 555 (1:500; ab150106, Abcam), anti-rabbit Alexa Fluor 488 (1:500; A-21206, Thermo fisher Scientific or ab150073, Abcam), and anti-rabbit Alexa Fluor 555 (1:500; ab150074, Abcam).

## Cell mapping and counting

For cell mapping and counting in the midbrain, coronal sections at 8.0-mm caudal from the bregma were studied. The rostrocaudal levels of the sections were validated by reference to appropriate landmarks, such as the cerebral aqueduct, periaqueductal gray, sensory roots of the trigeminal nerve, superior cerebellar peduncle, and superior and inferior colliculus nuclei. The distribution of CTb-labeled cells in the midbrain was investigated in eight rats. In three of the eight rats, distribution of ChAT-immunoreactive cells was also investigated. CTb-labeled and ChAT-immunoreactive cells were mapped onto digital images of original sections before they were transcribed to standard sections provided by Paxinos and Watson[62], again using the appropriate landmarks.

To count the number of Fos-immunoreactive cells in GFP or ChAT-immunoreactive populations in the MLR, one slice per rat that was randomly chosen from 2–4 successive slices. The immunoreactive cells were mapped onto digital images of original coronal sections as described above. The numbers of mapped cells in the CnF and PPTg, of which the extent was defined in accordance with the rat brain atlas[62] and by reference to the distribution of ChAT-immunoreactive cells as well as appropriate landmarks, were counted bilaterally in a double-blind manner.

The brain section images used in figures (Figs. 1a, c, e, i, 2a, e, 3a, 4a, 5a, Supplementary Figs. 2a, 6a, 7a, 10) were created by reference to the illustrations in Paxinos and Watson's atlas of the rat brain[62].

## Data analysis

Throughout in vivo optogenetic experiments, all analog measurements were digitized via the AD converter data acquisition system (PowerLab 8/30 or 8/35; ADInstruments), continuously displayed on a computer monitor, and digitally recorded at a 1-kHz sampling rate onto a hard disk using the LabChart software (v8.0 or 8.1.16; ADInstruments). MAP and HR were post hoc calculated beat-to-beat and resampled at 1 kHz. The overhead view of the experiment using conscious rats was continuously video-captured with a HD web camera (C920; Logicool) connected to a computer, and the video data were stored at a rate of 10 frames per s (fps).

In optogenetic experiments conducted in anesthetized or decerebrate rats, baseline values were obtained from averages for 15, 30, or 60 s before optogenetic interventions. Averaged values for 1 s immediately prior to each short period (0.5 s) of phostimulation period during the 1-min intermittent stimulation protocol were also determined to examine RSNA and VRNA responses to photostimulation. In experiments to study the effects of optogenetic stimulation via ChR2 activation in conscious rats, baseline values were obtained from 15-s averages before the onset of laser illumination. In experiments to study the effect of optogenetic inhibition via iChloC activation in conscious rats, mean values during the 2-s period immediately prior to laser illumination were determined as the baseline. Baseline values are presented in Supplementary Tables 1–14.

To quantify the RSNA and VRNA responses to photostimulation, after full-wave-rectified signals of these peripheral nerve activities, as well as background noise signals, were obtained, the noise component for the RSNA or VRNA was subtracted from the rectified signal. RSNA and VRNA responses to optogenetic interventions were quantified as relative changes from the baseline values, which were denoted as 100%. Additional procedures, i.e., superimposing and averaging analysis, were conducted to quantify RSNA and VRNA in response to a 0.5-s period of photostimulation during a 1-min intermittent stimulation protocol. Relative changes in RSNA or VRNA from the baseline were resampled at 1 kHz in response to each period of 0.5-s photostimulation, before being superimposed on one another and averaged at the time point (Fig. 2c). AUC values of the averaged RSNA changes were also calculated as an index of RSNA responses to photostimulation by integrating the increases in RSNA changes from the baseline during the time period following superimposing and averaging analysis (Fig. 2c).

Videos were recorded at 10 fps during experiments to study the effect of optogenetic stimulation in conscious rats; these were analyzed to calculate the locomotor speed using a video tracking system in accordance with its manual (SMART ver. 3.0; Panlab). To determine the contour of the rat in each image in a video sequence, an image of the experimental area (circular track) without a rat was used as reference and compared with any of the video images during the experiment that contained the rat. The difference between both images was then detected by the system. By consecutively detecting the center of rat's mass in each image, tracking of movement was achieved over 200 ms. Subsequently, the instantaneous rat speed in the circular track was calculated without smoothing. Videos recorded during

experiments to study the effect of optogenetic inhibition in conscious rats were used to validate the WRR values calculated using the above-described method.

Data were excluded from results in cases where injections missed the targeted region, the tips of optic fibers were inadequately located, disconnection of telemetry signals occurred during optogenetic stimulation for more than 1 s, or rats were not willing to voluntarily run on the wheel.

### Statistical analysis

All measurements were exported in Excel (Microsoft). All statistical analyses were performed in SigmaPlot 14.0 (Systat Software, Inc). For comparisons between independent groups, data were analyzed by Welch's $t$-test if they were normally distributed (assessed using the Shapiro–Wilk test) or by a Mann–Whitney U test if they were not normally distributed. For paired samples between trials, data were analyzed by paired $t$-tests. For repeated samples among trials or time course changes, data were analyzed by one-way repeated measures ANOVA (RM ANOVA) or by Friedman one-way RM ANOVA by rank where assumptions of normality (Shapiro–Wilk test) and/or equal variance (Brown–Forsythe test) were not met in ANOVA. For repeated samples among trials between independent groups, data were analyzed by two-way RM ANOVA. If appropriate, these ANOVA procedures were followed by Dunnett's, Holm–Sidak's, or Tukey's post hoc test. All tests were two-sided. Tests used to present each graph in figures and other statistic information including $F/t/\chi^2$ statistic values and degrees of freedom is presented in Supplementary Table 15. $P < 0.05$ was considered statistically significant. Data are expressed as means with standard errors (SEMs).

### Reporting summary

Further information on research design is available in the Nature Research Reporting Summary linked to this article.

## Data availability

All data associated with this study are provided in this published article and its supplementary information files. Source data are provided with this paper.

## Code availability

All code used to analyze sympathetic and cardiovascular responses is available on OSF: https://osf.io/tuhgm/.

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

## Acknowledgements

This research was supported by a JSPS KAKENHI Grant-in-Aid for Scientific Research (B) (21H03321) (S.K.); a JSPS KAKENHI Grant-in-Aid for Scientific Research (B) (18H03151) (S.K.); a JSPS KAKENHI Grant-in-Aid for Young Scientists (A) (15H05367) (S.K.)**;** a JSPS KAKENHI Grant-in-Aid for Challenging Research (Exploratory) (20K21759) (S.K.); a JSPS KAKENHI Grant-in-Aid for Scientific Research (C) (22K06470) (N.Ka.); a JSPS KAKENHI Grant-in-Aid for Scientific Research (C) (19K06954) (N.Ka.); a JSPS KAKENHI Grant-in-Aid for Scientific Research (B) (20H03418) (K.N.); by AMED (JP21wm0525002 to N.Ka.; JP21gm5010002 to K.N.); by JST Moonshot R&D (JPMJMS2023 to K.N.); and grants from the Takeda Science Foundation (S.K.). We thank Yui Yamane, Koji Maruyama, Atsuki Otake, Yumi Ono, and Eri Hanai for assistance with experiments and double-blinded manual cell counting, as well as the Tottori Bio Frontier managed by Tottori Prefecture, Japan, for the use of the confocal microscopy system.

## Author contributions

S.K. conceived the study. S.K. and N.Ku designed experiments with input from N.Ka. and K.N. S.K., N.Ka., K.N., and T.W. prepared study materials. N.Ku. and S.K. performed experiments and analyzed data. S.K., N.Ku., and E.N. prepared figures. S.K., N.Ku., and K.N. discussed data. S.K. and K.N. wrote the manuscript with input from all co-authors.

## Competing interests

The authors declare no competing interests.
