## [Peer Review File · Nature Communications]

A brainstem monosynaptic excitatory pathway that drives locomotor activities and sympathetic cardiovascular responsesREVIEWER COMMENTS

Reviewer #1 (Remarks to the Author):

This is a nice study that was technically challenging, and aimed at understanding some of the key sub-cortical neural circuits underpinning the cardiovascular response to exercise.

The authors provide compelling evidence using a combination of optogenetics, in vivo recordings and immunohistochemistry to support a monosynaptic pathway from the MLR to the RVLM controlling sympathetic outflow.

Areas for improvement.

Throughout the manuscript the authors loosely use the work 'central command' to underpin the importance of their findings. What has been studied are sub-cortical pathways that form a part of a relay circuit for autonomic output coupled to the motor system. A clearer introduction is required to make this point. Importantly, the MLR is not 'central command', but a sub-cortical command that is linked to higher centres during volitional movement.

The authors correctly reference (1,2) the early concept of cortical irradiation, but then need to introduce the concept of central command, which came from the classic paper of Goodwin et al. <https://doi.org/10.1113/jphysiol.1972.sp009979> 1972 J Physiol. Then reference Eldridge et al in 1981 who provided the critical link to the MLR in decorticate locomoting cats and showed the CV response occurred independent of afferent feedback <https://www.science.org/doi/abs/10.1126/science.7466362>. These two studies must be acknowledged early on as they set the framework for the current manuscript.

The next key historical point is to then introduce ref 42, which showed the role of stimulating the STN in humans with the associated CV response. A link can then be made to other mid areas that have been postulated to be key autonomic sites. In particular reference should be made to the PAG, which is not touched on given the strong expression you see in Fig 1C for MLR projections. The PAG has been extensively studied and shown in humans to be a major mid brain circuit underpinning sympathetic CV outflow (see Pain 2006 124(3):349-59 DOI: 10.1016/j.pain.2006.05.005; & ref 8; Hypertension 2014 May;63(5):1000-10. doi: 10.1161/HYPERTENSIONAHA.113.02970.].

Would encourage the authors to read the first part of this review

[J Physiol 2014 Feb 1;592(3):433-44.doi: 10.1113/jphysiol.2013.261586], which provides the historical context of discovery of key circuits involved in the the cardio-respiratory response to exercise.

To improve the scholarship of the discussion reference needs to be made to other mid circuits coupled to MRL-RVLM. A statement also needs to be made about the activation of the SMA/PMA and then motor cortex, which initiates the movement that is coupled to parallel activation of SNS. Citing the Asmussen muscle paralysis and Gandevia studies from the 1960's and 1990's will help the authors here provide overall context of the circuits, ie central command can work independently of movement feedback. Then focus on the MLR-RVLM link. A summary diagram would help readers here.

Some acknowledgement needs to be made about the limitations of the current methods used. In particular decerebrate preparations remove key neural pathways involved in volitional activity where thalamic inhibition is lost resulting in overactive mid-brain and brain stem activity.

In Figs 2 and 3 during opto stimulation it is surprising the HR did not increase, whereas RSNA and ABP increased Does this suggest your pathway is more involved in vascular control and not cardiac control during exercise? Comment?

In figs 4&5 where locomotion occurs you see the full CV response suggesting the movement itself has influenced the HR responses. Please comment?

Reviewer #2 (Remarks to the Author):

The manuscript by Koba and colleagues describes the results of experimental studies conducted in rats and aimed to test the hypothesis that neurons of the mesencephalic locomotor region (MLR) of the midbrain are responsible for coordination of locomotor activity and sympathetic cardiovascular responses during exercise. The study is expertly performed, the manuscript is well written and beautifully illustrated. Experiments involving tracing and optogenetic excitation are exemplary and support the study hypothesis. However, the results of the experiments involving optogenetic inhibition of MLR neurons are less convincing because of the reasons outlined below. In the opinion of this reviewer these experiments are essential to determine the role played by MLR in coordination of locomotor activity and sympathetic responses during exercise.

1. The efficacy of optogenetic inhibition of MLR neurons expressing iChloC needs to be demonstrated. iChloC channel-mediated inhibition had been characterised in the preceding studies but in the opinion of this reviewer the authors need to confirm that this approach can effectively inhibit the MLR neurons. This can be done using extracellular recordings in vivo or in slices.

2. I am not sure about the appropriateness of experimental design of these experiments. The authors say that "During spontaneous running on the wheel or at rest on the cage floor, the MLR was bilaterally illuminated for 2 s with 50-ms pulsed laser (10 mW at 10 Hz). In each rat, experiments were performed over 2–4 days, in which the laser pulse series was totally given 9–17 trials during running and 5–15 trials at rest". Respectfully, I did not find the data presented in Fig 5d-e convincing. The authors are showing a very short stretches of the recordings and it is impossible to evaluate the effect of exercise/locomotion and the kinetics of the response recovery after the light delivery to the MLR. Would it be more convincing to transduce MLR neurons with an inhibitory DREADD (for example), inhibit the neurons for a longer period of time and then give the animals an opportunity to exercise. This experimental paradigm would ultimately reveal the functional role of this neuronal group. The authors hypothesis will be supported if MLR inhibition leads to a blockade or reduction of locomotor and sympathetic responses to exercise.

3. Title: "A brainstem circuit coordinating locomotor activities and sympathetic cardiovascular responses". I am not sure that the authors study a "circuit"; "monosynaptic excitatory pathway" would be a better term. Also, I am not sure that the authors provide evidence of "coordination" between locomotor activities and sympathetic cardiovascular response by the MLR neurons. To make this conclusion the authors need to show evidence that in condition of MLR inhibition these activities are no longer coordinated in response to exercise. Perhaps "driving" (or "responsible for") would be a better term.

Responses to reviewers' comments

We sincerely thank you for supportive comments as well as constructive suggestions to improve our work. We have conducted additional experiments and revised the manuscript accordingly. In this letter, we have color-coded our responses in black whereas your original comments are in blue. The revised parts in the manuscript are shown in red.

To Reviewer #1:

This is a nice study that was technically challenging, and aimed at understanding some of the key sub-cortical neural circuits underpinning the cardiovascular response to exercise.

The authors provide compelling evidence using a combination of optogenetics, in vivo recordings and immunohistochemistry to support a monosynaptic pathway from the MLR to the RVLM controlling sympathetic outflow.

We thank you very much for positive evaluation on our work. To strengthen the manuscript, we have made revisions of the manuscript by following your comments. We express our greatest gratitude to you for providing us crucial comments to improve our manuscript.

Areas for improvement.

Throughout the manuscript the authors loosely use the work 'central command' to underpin the importance of their findings. What has been studied are sub-cortical pathways that form a part of a relay circuit for autonomic output coupled to the motor system. A clearer introduction is required to make this point. Importantly, the MLR is not 'central command', but a sub-cortical command that is linked to higher centres during volitional movement.

We agree with this point. According to your comment, we have made revisions throughout the manuscript to clearly demonstrate that the MLR→RVLM neurons constitute a subcortical pathway that relays central command signals (Page 2, Line 28; Page 4, Line 69; Page 5, Line 84; Page 14, Line 300).

The authors correctly reference (1,2) the early concept of cortical irradiation, but then need to introduce the concept of central command, which came from the classic paper of Goodwin et al. <https://doi.org/10.1113/jphysiol.1972.sp009979> 1972 J Physiol. Then reference Eldridge et al in 1981 who provided the critical link to the MLR in decorticate locomoting cats and showed the CV response occurred independent of afferent feedback <https://www.science.org/doi/abs/10.1126/science.7466362>. These two studies must be acknowledged early on as they set the framework for the current manuscript.

We agree with these comments. In the Introduction of the revised manuscript, we have demonstrated that the concept of central command has first come from observations by Goodwin *et al* (Page 3, Line 40). We now also state that central command is coupled to activation of sympathetic nervous system independently of movement feedback as revealed by Eldridge *et al* (Page 3, Line 43).

The next key historical point is to then introduce ref 42, which showed the role of stimulating the STN in humans with the associated CV response. A link can then be made to other mid areas that have been postulated to be key autonomic sites. In particular reference should be made to the PAG, which is not touched on given the strong expression you see in Fig 1C for MLR projections. The PAG has been extensively studied and shown in humans to be a major mid brain circuit underpinning sympathetic CV outflow (see Pain 2006 124(3):349-59 DOI: 10.1016/j.pain.2006.05.005; & ref 8; Hypertension 2014 May;63(5):1000-10. doi: 10.1161/HYPERTENSIONAHA.113.02970.).

We agree with the importance of the STN and PAG studies the reviewer referred to and have now accordingly incorporated these papers and their findings into the Introduction (Page 3, Line 52). Moreover, the strong expression in RVLM-projecting PAG neurons (Fig. 1c) has now been mentioned in the Discussion (Page 15, Line 305).

Would encourage the authors to read the first part of this review

[J Physiol 2014 Feb 1;592(3):433-44.doi: 10.1113/jphysiol.2013.261586], which provides the historical context of discovery of key circuits involved in the cardio-respiratory response to exercise.

Thank you very much for this comment. We have appreciated this review article, which was a guidepost to conceive the project, and has now been cited in appropriate positions (Page 4, Line 55; Page 15, Line 307).

To improve the scholarship of the discussion reference needs to be made to other mid circuits coupled to MRL-RVLM. A statement also needs to be made about the activation of the SMA/PMA and then motor cortex, which initiates the movement that is coupled to parallel activation of SNS. Citing the Asmussen muscle paralysis and Gandevia studies from the 1960's and 1990's will help the authors here provide overall context of the circuits, ie central command can work independently of movement feedback. Then focus on the MLR-RVLM link. A summary diagram would help readers here.

We appreciate these comments to improve our manuscript. In the Discussion, we have discussed the involvement of the PAG that should be coupled to the MLR-RVLM pathway (Page 15, Line 5). We have also stated that activation of SMA/PMA and motor cortex elicits the movement that is coupled to sympathoexcitation by citing the findings by Silber *et al* (2000) (Page 18, Line 380). Moreover, we have documented that central command can work independently of movement feedback by citing findings by Asmussen *et al.* (1963) and Gandevia *et al.* (1992). We have considered, nevertheless, that this concept should be stated earlier in Introduction along with the cat study by Eldridge and coworkers (1981). Therefore, these references have been cited in Introduction (Page 3, Line 43).

We have made a summary diagram of the central circuit mechanism for central command engaged for locomotion or running exercise, which has now been presented as Extended Data Fig. 10. We believe that this diagram will help readers find not only the core result of this study but also unsolved questions to reveal the whole picture of the central circuitry mechanism for central command.

Some acknowledgement needs to be made about the limitations of the current methods used. In particular decerebrate preparations remove key neural pathways involved in volitional activity where thalamic inhibition is lost resulting in overactive mid-brain and brain stem activity.

Agreeing with this point, we have stated the limitation of the present decerebrate rat preparation (Page 8, Line 166).

In Figs 2 and 3 during opto stimulation it is surprising the HR did not increase, whereas RSNA and ABP increased Does this suggest your pathway is more involved in vascular control and not cardiac control during exercise? Comment?

The answer is yes. The RVLM predominantly control sympathetic vasomotor activity, rather than cardiac functions (Guyenet. *Nat Rev Neurosci* 7:335–346, 2006). It was also shown that optogenetic inhibition of RVLM C1 neurons elicits tachycardia in resting, conscious rats (Wenker *et al.* *J Neurosci* 37: 4565-4583, 2017). This observation suggests that exercise tachycardia may be principally mediated by other circuitry mechanisms than excited MLR→RVLM neurons. This suggestion is supported by our data showing no effect of optogenetic inhibition of the MLR→RVLM pathway on heart rate during voluntary running exercise.

In figs 4&5 where locomotion occurs you see the full CV response suggesting the movement itself has influenced the HR responses. Please comment?

We agree with your interpretation that the movement itself during exercise is likely involved in the tachycardia. Activation of the exercise pressor reflex due to the movement should be involved in the HR response (Mitchell *et al.* *Annu. Rev. Physiol.* 45, 229–242, 1983). It is also possible that MLR→RVLM pathway activation subsequently stimulates other central circuits that in turn elicit cardiovascular responses. For example, the MLR regulates cortical state in parallel with locomotion (Lee *et al.* *Neuron* 83; 455–466, 2014). Other circuitries than MLR→RVLM pathway, of which activation was triggered by stimulation of MLR→RVLM pathway, may also explain the tachycardiac response. We have stated these discussions in the revised manuscript (Page 11, Line 223).

To Reviewer #2:

The manuscript by Koba and colleagues describes the results of experimental studies conducted in rats and aimed to test the hypothesis that neurons of the mesencephalic locomotor region (MLR)

of the midbrain are responsible for coordination of locomotor activity and sympathetic cardiovascular responses during exercise. The study is expertly performed, the manuscript is well written and beautifully illustrated. Experiments involving tracing and optogenetic excitation are exemplary and support the study hypothesis. However, the results of the experiments involving optogenetic inhibition of MLR neurons are less convincing because of the reasons outlined below. In the opinion of this reviewer these experiments are essential to determine the role played by MLR in coordination of locomotor activity and sympathetic responses during exercise.

We appreciate your evaluation on our manuscript and positive comments. We also understand your concerns. To strengthen the study findings, we have performed additional experiments and analyses and made revisions of the manuscript. We express our greatest gratitude to your comments, that have made the revised manuscript be much improved.

1. The efficacy of optogenetic inhibition of MLR neurons expressing iChloC needs to be demonstrated. iChloC channel-mediated inhibition had been characterised in the preceding studies but in the opinion of this reviewer the authors need to confirm that this approach can effectively inhibit the MLR neurons. This can be done using extracellular recordings in vivo or in slices.

We agree with this point. Accordingly, we have performed additional experiments, thereby confirmed that iChloC channel activation inhibits MLR→RVLM neurons as below.

We addressed this issue by performing immunohistochemistry to detect Fos expression, a marker for neuronal activation, in MLR→RVLM neurons. We prepared three groups of SD rats (8~16 weeks old, both sex): “control rats” that received MLR injection of AAV-EF1a-DIO-eYFP, “ChR2 rats” that received MLR injection of AAV-EF1a-DIO-ChR2-eYFP, and “ChR2&iChloC rats” that received MLR injection of AAV-EF1a-DIO-ChR2-eYFP and AAV-EF1a-DIO-iChloC-mcherry. All rats also received RVLM injection of AAVrg-pgk-cre. More than 3 weeks after injections, the MLR was bilaterally and intermittently (10-s on and 10-s off, 90 times) illuminated for 30 min with 50-ms pulsed blue laser (10 mW, 10 Hz) under anesthesia with a mixture of urethane and alpha-chloralose. Forty-five minutes after the offset of illumination, the brain was taken after perfusion and fixation with 4% paraformaldehyde. Coronal brain slices including the MLR were prepared. Unfortunately, our triple immunohistochemistry to simultaneously stain Fos, eYFP, and mcherry was failed

because several antibodies we obtained from commercial companies did not work. Thus, we conducted two sets of double immunohistochemical staining of Fos & eYFP and of eYFP & mcherry.

We found that iChloC was colocalized in $81 \pm 8\%$ of Chr2-expressing MLR-RVLM neurons of Chr2&iChloC rats as shown by overlapping RFP (mcherry) immunoreactivities with GFP (eYFP) immunoreactivities. Fos expression was induced in MLR→RVLM neurons by Chr2-mediated optogenetic activation of their cell bodies in the MLR, and this induced Fos expression was suppressed by simultaneous photoactivation of iChloC in these neurons. These findings demonstrate that excitation of MLR→RVLM neurons due to photostimulation of Chr2 is effectively inhibited by co-photostimulation of iChloC.

The results have now been stated in the revised manuscript (Page 12, Line 253) and presented in Extended Data Figure 7. The procedures for this experiment were also stated in the revised manuscript (Page 33, Line 708).

2. I am not sure about the appropriateness of experimental design of these experiments. The authors say that "During spontaneous running on the wheel or at rest on the cage floor, the MLR was bilaterally illuminated for 2 s with 50-ms pulsed laser (10 mW at 10 Hz). In each rat, experiments were performed over 2–4 days, in which the laser pulse series was totally given 9–17 trials during running and 5–15 trials at rest". Respectfully, I did not find the data presented in Fig 5d-e convincing. The authors are showing a very short stretches of the recordings and it is impossible to evaluate the effect of exercise/locomotion and the kinetics of the response recovery after the light delivery to the MLR. Would it be more convincing to transduce MLR neurons with an inhibitory DREADD (for example), inhibit the neurons for a longer period of time and then give the animals an opportunity to exercise. This experimental paradigm would ultimately reveal the functional role of this neuronal group. The authors hypothesis will be supported if MLR inhibition leads to a blockade or reduction of locomotor and sympathetic responses to exercise.

We understand your concern. Your suggestion to inhibit the neurons for a longer period is greatly appreciated. We conducted another set of experiments in which the effect of genetic ablation of MLR→RVLM neurons on locomotion and cardiovascular responses during running exercise elicited by MLR stimulation was tested (we did not employ DREADD

mainly because it did not work in our previous studies using rats). In “control” rats, we injected to the RVLM with a retrograde AAV encoding Cre and to the MLR with an anterograde AAV encoding Cre-independent ChR2-eYFP. In “ablate” rats, we additionally injected to the MLR with an anterograde AAV encoding Cre-dependent caspase. Therefore, in “ablate” rats, apoptosis of Cre-positive, RVLM-projecting MLR neurons should have been mediated by Cre-dependent expression of caspase. Unfortunately, however, we failed to establish the procedure to genetically ablate rat MLR→RVLM neurons because Cre-positive, RVLM-projecting MLR neurons were sufficiently present in “ablate” rats as seen in “control” rats (please see *Figure for Review*). Therefore, we have revised the manuscript without the “ablation” experiment but included new information obtained from additional analyses explained in the subsequent paragraphs.

Regarding the appropriateness of experimental design, we would like to note that the experimental design has been sophisticated to the best degree after multiple trials and errors in pilot experiments. As we employed voluntary wheel running, the rats could anytime either start, continue, or discontinue wheel running as they wanted. In other words, the voluntary running period in rats varied and was uncontrollable, that was 5~15 sec in many cases and sometimes longer than 15 s in male adult rats which were manipulated with telemeter and optical fibers. Moreover, the average running duration, which reflected the willingness of the rat to run, was also different among rats. Therefore, we supposed 10 sec as the assumed running period and determined 2-sec as the duration for MLR illumination during voluntary

running in order to minimize the effect of “voluntary discontinuation of running” on the locomotor and cardiovascular changes. If the period for optogenetic intervention was longer than 2-sec, the probability that the effect of voluntary discontinuation of running is included in the results would be increased. Consequently, we believe that the experimental design was the best to test the effect of optogenetic inhibition of MLR→RVLM neurons on locomotor activities and cardiovascular responses during voluntary running exercise.

To reinforce the convincingness of the results from our experiments, we conducted further analyses to quantitatively investigate the effect of optogenetic inhibition to change behavioral patterns during running and their association with cardiovascular changes. As stated in the revised manuscript (Page 13, Line 260), optogenetic interventions during voluntary wheel running were followed by various patterns of behavioral changes among trials in each rat. The behavioral pattern following MLR illumination included 1) continuing running on the wheel, 2) slowing running, 3) re-running after a pause or slowing, 4) standing on the wheel after stopping running, and 5) leaving the wheel for the cage floor. We looked through the original data and videos, thereby categorized the recovery-behavior into these five patterns. Then, we compared the cardiovascular responses as well as locomotor speed index (assessed by WRR) among the recovery-behavior groups.

We found that the probability that running was continued through the 5-s period after the onset of 2-s laser pulse series was significantly higher in eYFP-expressing controls than that in iChloC-mCherry-expressing rats, whereas the probabilities of other types of behavior with reduced locomotor activity (pause and slowing) during the 5-s period were significantly greater in iChloC rats than those in controls (Extended Data Fig. 8b). These results indicate that MLR→RVLM neuronal excitation is required for continuation of voluntary wheel running exercise. We further found that the locomotor activity reduction was associated with a decrease in blood pressure, but not heart rate; blood pressure changes in iChloC rats during optogenetic intervention and recovery displayed similar kinetics with WRR changes, irrespective of behavioral pattern (Extended Data Fig. 8c). The similar kinetics suggest that the prevented locomotor activities due to optogenetic inhibition are not a cause of blood pressure decreases or vice versa. Totally, the results obtained from these additional analyses further strengthened the notion that MLR→RVLM neurotransmission mediates the central

command signaling to simultaneously drive locomotor activities and sympathetic cardiovascular responses during voluntary running exercise. These results have now been reported in the revised manuscript (Page 13, Line 258) and in Extended Data Figure 8.

3. Title: "A brainstem circuit coordinating locomotor activities and sympathetic cardiovascular responses". I am not sure that the authors study a "circuit"; "monosynaptic excitatory pathway" would be a better term. Also, I am not sure that the authors provide evidence of "coordination" between locomotor activities and sympathetic cardiovascular response by the MLR neurons. To make this conclusion the authors need to show evidence that in condition of MLR inhibition these activities are no longer coordinated in response to exercise. Perhaps "driving" (or "responsible for") would be a better term.

We agree with these points and thank you very much for this correction. Accordingly, we have revised the title. The title is "A brainstem monosynaptic excitatory pathway that drives locomotor activities and sympathetic cardiovascular responses".

REVIEWERS' COMMENTS

Reviewer #1 (Remarks to the Author):

The authors have done an excellent job in revising the manuscript and improving the overall scholarship of the presentation. The introduction now accurately reflects the historical literature and new results significantly add to the field of study.

Reviewer #2 (Remarks to the Author):

The authors satisfactorily addressed my comments and revised the manuscript accordingly. However, in my opinion the authors should still consider improving the data presentation in Fig 5. As I've argued in my review of the original submission, the authors are showing a very short stretches of the recordings from which it is impossible to evaluate the effect of exercise/locomotion on cardiovascular variables. I fully appreciate the complexity of these experiments and the fact that the duration of voluntary running periods was variable, but in my opinion it would be helpful to see the individual data profiles of the running episodes with 5-10 s baseline and 5-10 s post-exercise recovery. I also have one additional minor point to make: increase in heart rate is a hallmark of exercise and sympathetic activation contributes significantly to exercise-induced tachycardia; perhaps the authors may wish to discuss why activation of the neuronal pathway under investigation has no effect on heart rate in anaesthetised and decerebrate preparations (in freely behaving animals [Fig 4] increases in HR might be secondary to the locomotor response).

Responses to reviewers' comments

We sincerely thank you for supportive comments as well as constructive suggestions to improve our work. Following suggestions and the editorial requests, we have conducted additional analyses and revised the manuscript one last time. The revised parts in the manuscript are shown in red. In this letter, we have color-coded our responses in black whereas your original comments are in blue.

To Reviewer #1:

The authors have done an excellent job in revising the manuscript and improving the overall scholarship of the presentation. The introduction now accurately reflects the historical literature and new results significantly add to the field of study.

We cordially appreciate your positive evaluation towards our manuscript.

To Reviewer #2:

The authors satisfactorily addressed my comments and revised the manuscript accordingly. However, in my opinion the authors should still consider improving the data presentation in Fig 5. As I've argued in my review of the original submission, the authors are showing a very short stretches of the recordings from which it is impossible to evaluate the effect of exercise/locomotion on cardiovascular variables. I fully appreciate the complexity of these experiments and the fact that the duration of voluntary running periods was variable, but in my opinion it would be helpful to see the individual data profiles of the running episodes with 5-10 s baseline and 5-10 s post-exercise recovery. I also have one additional minor point to make: increase in heart rate is a hallmark of exercise and sympathetic activation contributes significantly to exercise-induced tachycardia; perhaps the authors may wish to discuss why activation of the neuronal pathway under investigation has no effect on heart rate in anaesthetised and decerebrate preparations (in freely behaving animals [Fig 4] increases in HR might be secondary to the locomotor response).

Thank you very much for your constructive comments to improve our work. We agree with your point, that led us to further conduct analyses to demonstrate data profiles of the running episodes with 5 s baseline and 5 s post-exercise recovery. The graphs generated after these additional analyses have now been presented in Figure 5e and Supplementary Fig 8a & 8c.

In the current manuscript, we have made statements to discuss why activation of MLR→RVLM pathway had no effect on heart rate in anesthetized and decerebrate preparations. We also agree with your

interpretation and have now mentioned that increases in HR by MLR→RVLM neuron stimulation might be secondary to the locomotor response (Page 11).

Again, we cordially appreciate your suggestions to improve the manuscript.